# Stress-induced RNA–chromatin interactions promote endothelial dysfunction

Riccardo Calandrelli [1,5], Lixia Xu[2,3,5], Yingjun Luo [3,5], Weixin Wu[1], Xiaochen Fan[1], Tri Nguyen[1], Chien-Ju Chen[1], Kiran Sriram[3,4], Xiaofang Tang[3], Andrew B. Burns [3], Rama Natarajan [3,4], Zhen Bouman Chen [3,4✉] & Sheng Zhong [1✉]

Chromatin-associated RNA (caRNA) has been proposed as a type of epigenomic modifier. Here, we test whether environmental stress can induce cellular dysfunction through modulating RNA-chromatin interactions. We induce endothelial cell (EC) dysfunction with high glucose and TNFα (H + T), that mimic the common stress in diabetes mellitus. We characterize the H + T-induced changes in gene expression by single cell (sc)RNA-seq, DNA interactions by Hi-C, and RNA-chromatin interactions by iMARGI. H + T induce interchromosomal RNA-chromatin interactions, particularly among the super enhancers. To test the causal relationship between H + T-induced RNA-chromatin interactions and the expression of EC dysfunction-related genes, we suppress the *LINC00607* RNA. This suppression attenuates the expression of *SERPINE1*, a critical pro-inflammatory and pro-fibrotic gene. Furthermore, the changes of the co-expression gene network between diabetic and healthy donor-derived ECs corroborate the H + T-induced RNA-chromatin interactions. Taken together, caRNA-mediated dysregulation of gene expression modulates EC dysfunction, a crucial mechanism underlying numerous diseases.

[1] Department of Bioengineering, University of California San Diego, 9500 Gilman Dr., La Jolla, CA 92093, USA. [2] Division of Nephrology, Guangdong Provincial People's Hospital, Guangdong Academy of Medical Sciences, Guangzhou, China. [3] Department of Diabetes Complications and Metabolism, Beckman Research Institute, City of Hope, 1500 Duarte Rd., Duarte, CA 91010, USA. [4] Irell and Manella Graduate School of Biological Sciences, City of Hope, 1500 Duarte Rd., Duarte, CA 91010, USA. [5] These author contributed equally: Riccardo Calandrelli, Lixia Xu, Yingjun Luo. ✉email: zhenchen@coh.org; szhong@ucsd.edu

Mammalian genomes are extensively transcribed to RNAs, and a portion of RNAs are physically associated with chromatins and thus are termed chromatin-associated RNAs (caRNAs)[1]. However, the structural and functional role of these caRNAs in 3D nuclear organization and transcriptional regulation remains unclear. Despite the increasing evidence supporting that caRNAs play important roles in regulating nuclear function and transcriptional output[2], most of these studies focused on individual caRNAs[3–5]. In contrast, there is little information on global RNA–chromosomal interactions on a genome-wide scale.

Recent technological developments have made it possible to assay DNA–DNA and RNA–chromatin interactions in situ in a genome-wide manner[6–11]. Among these tools, in situ mapping of RNA–genome interactome (iMARGI) enables all-RNA-versus-the-genome analyses that can simultaneously identify many caRNAs and their respective genomic interaction loci[7,8]. This feature helped to reveal a large number of caRNAs, including those attached to other chromosomes[7,12]. However, it is unknown whether these RNA–chromatin contacts are altered in a dynamic cellular process, and how these interactions impact transcriptional output and functional outcome in the context of health and disease remains unclear.

Endothelial cells (ECs) lining the interface between circulating blood and vascular wall are crucial for the vital function of every tissue and organ with blood perfusion. Many pathological conditions, including the epidemic diabetes that is associated with hyperglycemia and chronic inflammation, can cause EC dysfunction. During EC dysfunction, ECs undergo transcriptional changes that impair homeostatic function (e.g., nitric oxide production and angiogenesis), while inducing pro-inflammatory and pro-fibrotic responses. Although the importance of endothelial dysfunction has been well documented in many diseases[13], the underlying molecular mechanisms, particularly those involving changes in chromatin organization remain largely unknown. An earlier work has underscored the importance of super enhancers (SEs) in inflammatory transcription in ECs[14]. Our previous work suggests that an enhancer-derived long noncoding RNA (lncRNA) can promote the transcription of endothelial nitric oxide synthase (eNOS) through interchromosomal RNA–DNA interactions[5]. Given that many enhancers are actively transcribed into RNAs that remain bound to chromatins[15], it is curious what regulatory role these enhancer/SE-embedded caRNAs play in transcriptional regulation of EC dysfunction.

In this study, we leverage time-course iMARGI analysis together with time-course Hi-C and single-cell transcriptome analyses to interrogate how global RNA–chromosomal contacts change in EC dysfunction associated with diabetes, and to what extent these changes impact cell phenotype and function. We first established an in vitro system where we induce EC dysfunction encompassing a robust pro-inflammatory activation and endothelial–mesenchymal transition (EndoMT) phenotype. We then employed a combination of single-cell RNA sequencing (scRNA-seq), Hi-C, and iMARGI analysis to characterize temporal changes in the transcriptome, genomic interactome, and RNA–chromatin interactome. The dysfunctional ECs exhibit a large number of interchromosomal RNA–chromatin interactions, especially among the SEs from different chromosomes. These SEs overlap with key regulatory genes promoting multi-facets of EC dysfunction, including inflammation, extracellular matrix (ECM) remodeling, and EndoMT. Among the emergent interchromosomal interactions in dysfunctional ECs, we identified an interaction involving a SE on chromosome 2 overlapping LINC00607 (a long intergenic noncoding RNA with unknown function), and a SE on chromosome 7 overlapping SERPINE1/PAI-1 (plasminogen activator inhibitor, a crucial regulator in

endothelial dysfunction and many vascular diseases)[16]. Perturbing the RNA-chromatin contacts by LINC00607 knockdown leads to the suppression of SERPINE1 and other genes contributing to endothelial dysfunction, as well as attenuation of monocyte adhesion and EC senescence. Correlational analysis performed with scRNA-seq data from H + T-treated ECs and diabetic donor-derived ECs reveal that LINC00607 and SERPINE1 are co-expressed in the same single cells more often in the dysfunctional ECs than in healthy control ECs. Collectively, our data suggest that RNA–chromatin interactions contribute to transcriptional regulation during endothelial dysfunction, a biological process closely implicated in various diseases.

## Results

**High glucose and TNFα induce EC dysfunction.** We first established and characterized an in vitro model of endothelial dysfunction by subjecting human umbilical vein endothelial cells (HUVECs) to high glucose (HG, 25 mM D-glucose) and TNFα (5 ng/mL, to mimic inflammation; H + T) for 3 and 7 days (abbreviated as H + T Days 3 and 7). The control cells were kept in 25 mM mannitol (NM control), denoted as Day 0. This design is based on three premises: (1) hyperglycemia and chronic inflammation are two key culprits in diabetes to cause EC dysfunction[17,18]; (2) the prolonged and combined treatment would induce robust EC changes, encompassing eNOS suppression, pro-inflammatory activation, ECM remodeling, and EndoMT, in which ECs manifest a phenotypic transition into mesenchymal-like cells[19,20]; and (3) the time course will allow temporal mapping of EC changes in transcriptome, genomic interactions, and RNA–genome interactions.

We first characterized this in vitro EC dysfunction model at the single-cell transcriptome level. scRNA-seq revealed that ECs underwent dramatic transcriptional changes under H + T treatment, evident by three separate clusters on the t-distributed stochastic neighbor embedding (t-SNE) plot corresponding to three different time points, i.e., Day 0, H + T Day 3, and H + T Day 7 (Fig. 1b). However, principal component analysis (PCA) showed that ECs across three time points were not clustered separately (Fig. 1c), implying that ECs remain largely the same population, despite clear differences in transcriptional states. Differential expression analysis identified 457 genes between Day 0 vs. Day 3 (p value < 9.72e−27, Wilcoxon test) and 826 genes between Day 0 vs. Day 7 (p value < 9.93e−115, Wilcoxon test). Among these differentially expressed (DE) genes, 181 are consistently upregulated and 173 consistently downregulated by H + T treatment (Supplementary Fig. 1).

Subsequent pathway enrichment analysis of DE genes demonstrated a significant enrichment of key pathways contributing to endothelial dysfunction, with a number of genes induced to promote inflammatory response (e.g., intercellular adhesion molecule (ICAM1), monocyte chemoattractant protein 1 (encoded by CCL2), PAI-1 (encoded by SERPINE1)), ECM remodeling (e.g., fibronectin (FN1) and collagens (COL4, COL5, and COL8)), and transforming growth factor (TGF-β) signaling and fibrotic pathways (e.g., TGFB1, TGFB2, SMAD3, and connective tissue growth factor (CTGF)), while eNOS, encoded by NOS3, and many EC function markers decreased (Fig. 1d, e and Supplementary Table 1). In addition to protein-coding genes, several lncRNAs were also detected as DE genes. These include the H + T-upregulated LINC00607, LINC01013, and LINC02154, and the downregulated LINC01235 (Fig. 1d, e).

scRNA-seq data also revealed the heterogeneity of EC transcriptomic changes. For example, the expression of EC hallmark gene eNOS was significantly decreased by H + T, evident not only by the reduced average mRNA level in single

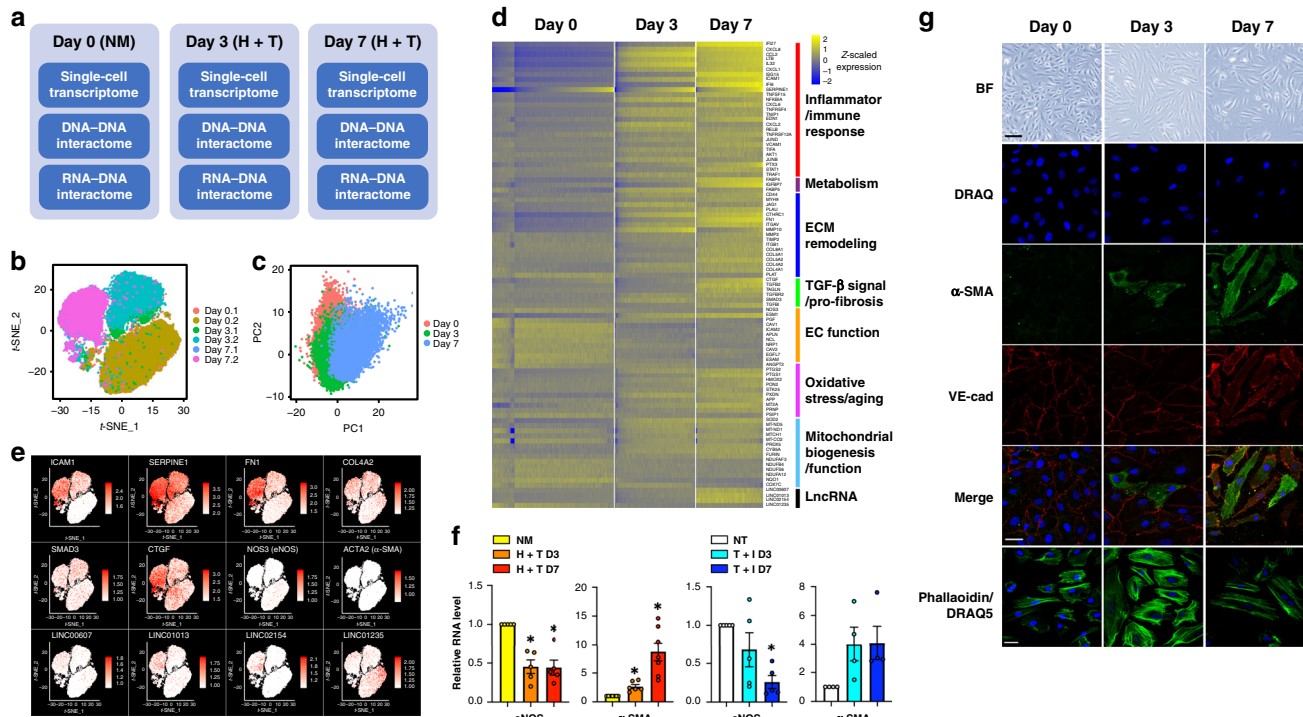

**Fig. 1 High glucose and TNFα induce a profound gene expression and phenotypic change. a** HUVECs were treated in biological duplicates as Day 0: 25 mM mannitol as normal glucose and osmolarity control (NM); Day 3: combined treatment consisting 25 mM D-glucose and 5 ng/mL TNFα (H + T) for 3 days; and Day 7: H + T treatment for 7 days. Each group of treated cells was subjected to single-cell RNA-seq (scRNA-seq), Hi-C, and iMARGI assays. **b** t-SNE plot of scRNA-seq (4000–15,000 cells per sample) showed clear separation by treatment condition into three distinct clusters. **c** Principal component analysis of scRNA-seq data: single cells are plotted in the first two PC space and are labeled in red (Day 0, i.e., NM), green (Day 3, i.e., 3-day H + T treatment), and blue (Day 7, i.e., 7-day H + T treatment). **d** Expression heatmap (z-scaled) of top DE genes in single ECs grouped into functional pathways. Cells were ordered by increasing *SERPINE1* expression (per each sample separately) and binned per 100 cells for the analysis. A total of 269 bins in Day 0, 177 bins in Day 3, and 148 bins in Day 7. **e** t-SNE plots of the expression level of selected genes in each single cell across the time course. The RNA levels are represented by log-normalized unique molecular identifier counts. **f** mRNA levels of eNOS and α-SMA in NM vs. H + T-treated HUVECs and cells untreated (NT) or treated with TGF-β (10 ng/mL) and IL-1β (5 ng/mL; T + I) for 3 or 7 days. The respective control was set as 1. Relative eNOS level: data represent mean ± SEM from five independent experiments; relative α-SMA level in H + T treatment: data represent mean ± SEM from seven independent experiments; relative α-SMA level in T + I treatment: data represent mean ± SEM from four independent experiments. *$P = 0.0067, 0.0087,$ 0.0057, and 0.0017 from left to right based on ANOVA with Bonferroni as post hoc test. **g** Cell morphology under bright field (BF), immunofluorescent staining of α-SMA, and VE-cadherin (VE-cad), phalloidin staining of cytoskeleton, and DRAQ5 (DRAQ) staining of the nuclei. Representative images from five independent experiments are shown. Scale bar of BF = 100 μm; scale bars of (immuno)fluorescent staining = 50 μm. Source data are provided as a Source data file.

ECs, but also by the lowered percentage of ECs that express eNOS, following a time-dependent manner. In the same time course, pro-inflammatory and pro-fibrotic genes (e.g., *ICAM1, FN1*, and *SERPINE1*) were increased in mRNA levels in single ECs and in percentage of ECs with positive expression across time (Fig. 1d, e). We also observed varied patterns of transcriptional changes among DE genes engaged in different molecular pathways and cellular functions. The most robust induction was in the pro-inflammatory genes. For instance, *ICAM1* was detected in <2% of control ECs at Day 0, in 68% of ECs at Day 3, and 89% of ECs at Day 7. Similar but less drastic dynamics were observed for genes involved in ECM organization and remodeling. For example, *FN1* was expressed in 36% of control ECs, which increased to 73% by Day 3 and then to 95% by Day 7. In contrast, the mesenchymal marker smooth muscle cell actin (α-SMA, encoded by *ACTA2*) was induced in a much slower pattern, i.e., from 0.2% of control ECs to 1.4% of cells after 7 days of H + T treatment (Fig. 1e). These results suggest a time-dependent signaling cascade initiated by a strong inflammatory response, which relays to substantial ECM remodeling and eventually perpetuates TGF-β signaling and EndoMT.

To characterize the EC changes at the cellular level and confirm that H + T-treated cells undergo an EndoMT process, we verified the expression of eNOS and α-SMA in bulk ECs using quantitative PCR (qPCR; Fig. 1f). As a positive control, we treated ECs with TGF-β and interleukin 1 beta (IL-1β), which has been demonstrated to induce EndoMT[21,22]. Both treatments caused apparent morphological changes in ECs, accompanied by suppression of eNOS and induction of α-SMA at mRNA levels, with H + T inducing a stronger EC morphological change and a higher induction of α-SMA (Fig. 1f, g and Supplementary Fig. 2). Consistently, α-SMA was also progressively increased at the protein level by H + T, as visualized by immunofluorescent staining, while VE-cadherin (VE-cad), an EC-specific membrane marker, remained expressed in ECs (Fig. 1g). The morphological change is also evident by phalloidin staining of cytoskeleton, which demonstrates that ECs with the typical classic cobblestone, began to transition into mesenchymal-like spindle shape by 3 days, which became more distinguished by 7 days (Fig. 1g). Following experiments, including Hi-C and iMARGI assays were performed with ECs treated by the same H + T condition in the same time course (i.e., Days 0, 3, and 7, Fig. 1a).

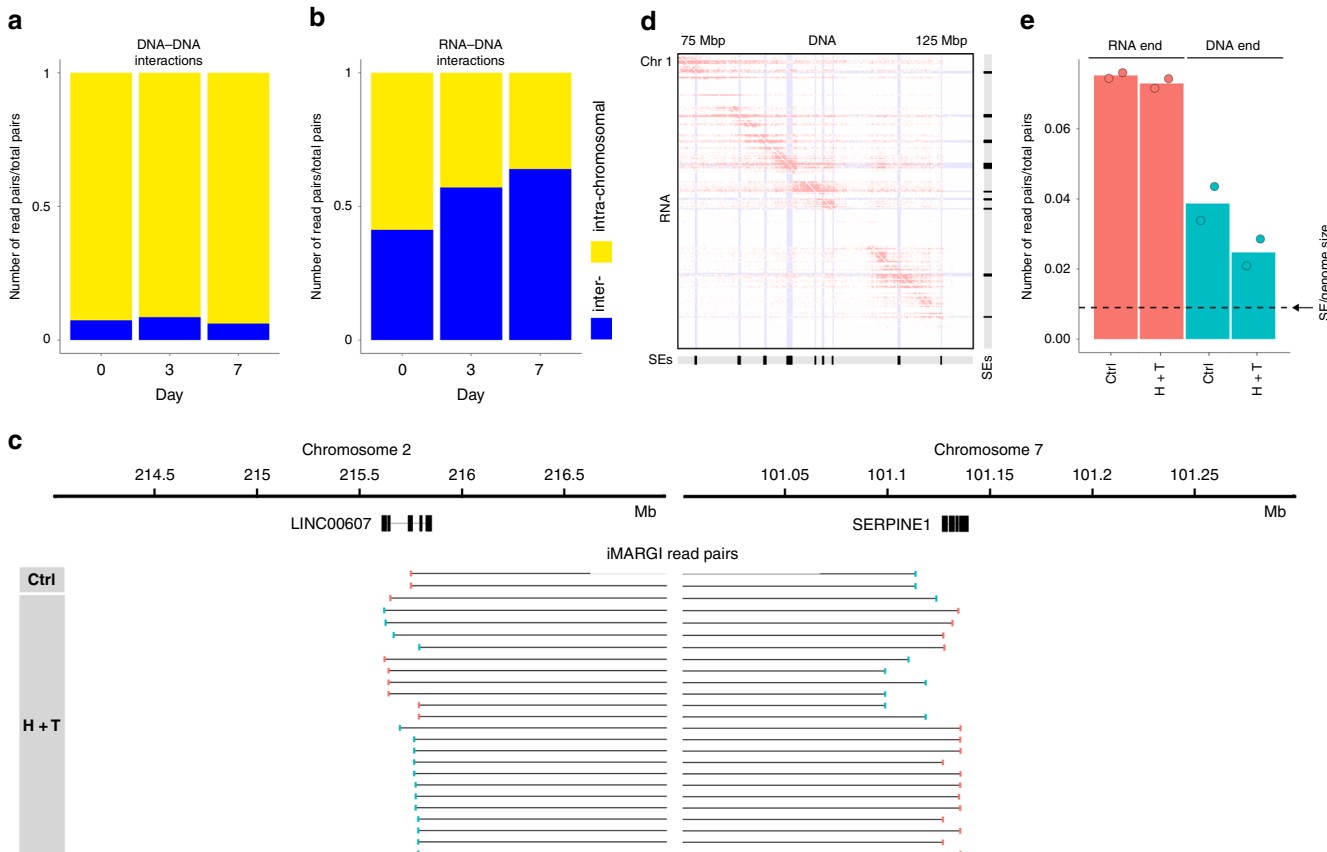

**Fig. 2 Overview of time-course Hi-C and iMARGI data. a**, **b** Proportions of intrachromosomal (yellow) and interchromosomal read pairs (blue) in Hi-C (**a**) and iMARGI data (**b**) at the three time points (columns). **c** An example of interchromosomal iMARGI read pairs mapped to chromosome 2 near *LINC00607* (left) and chromosome 7 near *SERPINE1* (right). The RNA end (pink) and the DNA end (green) of each read pair is linked by a horizontal line. **d** Examples of overlapping iMARGI read pairs on a contact matrix from the RNA end (rows) to the DNA end (columns) with SEs (marked in light blue and in SE tracks) in control (Day 0) ECs. Genome region: chr1:75,000,0000–chr1:125,000,000. Resolution = 200 kb. **e** Proportions of iMARGI read pairs with the RNA ends (pink) or the DNA ends (green) in Day 0 (Ctrl) and Days 3 and 7 (H + T) ECs mapped to HUVEC SEs. Dotted line: the relative size of HUVEC SEs compared to the size of the genome. Source data are provided as a Source data file.

**Lack of significant changes in 3D genome organization in ECs.** We investigated whether H + T induced any changes in the 3D genome structure by using in situ Hi-C. Among 164,585,295 uniquely mapped read pairs on average from each time point, the proportions of intrachromosomal read pairs were 92.6%, 91.4%, and 93.8% at Days 0, 3, and 7, respectively, which did not exhibit a trend of change over time (Fig. 2a). The Hi-C-derived A/B compartments and topologically associated domains (TADs) did not exhibit notable changes either (Supplementary Fig. 3a, b). To assess the degree of TAD changes at the genome scale, we calculated the Measure of Concordance (MoC)[23] between every two time points (Supplementary Fig. 3c). The MoCs of the three pairwise comparisons were all categorized as highly concordant based on the cutoff of MoC > 0.75 as recommended[23]. These data suggest that H + T did not significantly perturb the major 3D genome features. The lack of genome-wide changes in 3D genome structure is consistent with the observation that the single-cell transcriptomes remained a single cluster in the same time course (Fig. 1c).

**Changes of RNA–chromatin interactions in dysfunctional ECs.** Next, we asked whether H + T induces any changes in RNA–chromatin interactions. We subjected ECs of the same H + T treatment to genome-wide RNA–chromatin interaction profiling by iMARGI. To account for biological variability, we repeated the experiment and generated a second iMARGI dataset from

Day 0 to Day 7 ECs (Supplementary Table 2). We describe two types of robustness analyses within the results, based on (1) comparing the H + T-induced changes between Days 7 and 0, each with two biological replicates, and (2) merging and splitting the two time points after treatment (Days 3 and 7) for comparison with Day 0.

In the control ECs (Day 0), 34.4% of the uniquely mapped iMARGI read pairs were interchromosomal read pairs, which is on a comparable scale but smaller than the previously reported 52% in human embryonic kidney cells[7]. After treatment, interchromosomal read pairs increased to 62.7% (Days 3 and 7 combined; p value = 1e−4, d.f. = 1, chi-square test). Consistently, when we separately analyzed the three time points, the trend of increase from Day 0 to Day 3 persisted through Day 3 to Day 7 (Fig. 2b, c). Thus, H + T treatment induced interchromosomal RNA–chromatin interactions in ECs. Such a pronounced change is in contrast to the lack of 3D genome structural changes (Fig. 2a and Supplementary Fig. 3), begging the question whether the interchromosomal RNA–chromatin interaction changes contribute to the observed transcriptome changes in the same time course (Fig. 1).

**Enrichment of SE-derived RNAs in chromatin-associated RNAs.** We asked whether the RNA transcripts from intergenic regulatory sequences, such as enhancer RNAs (eRNAs) are a major component of caRNAs. To this end, we obtained the

coordinates of 63,758 HUVEC enhancers from EnhancerAtlas[24] and 912 HUVEC SEs from dbSUPER[25]. Read 1 and read 2 of each iMARGI read pair are referred to the RNA end (read 1) and the DNA end (read 2), because they are converted from the RNA and the DNA[7]. A total of 7,704,090 (10%) iMARGI read pairs had their RNA ends mapped to enhancer regions. Compared to the total length of enhancers in HUVEC (120,382,426 bp, ~3.9% of the genome), eRNA–DNA read pairs are enriched in iMARGI data (odds ratio = 2.7, p value < 2.2e−16, d.f. = 1, chi-square test; Supplementary Fig. 4a), suggesting eRNA–chromatin interactions as a major component of RNA–chromatin interactions. Furthermore, 5,936,114 (7.6%) of iMARGI read pairs had their RNA ends mapped to SE regions, whereas the total length of SEs (28,277,698 bp) only accounts for 0.9% of the genome size (odds ratio = 9, p value < 2.2e−16, d.f. = 1, chi-square test; Fig. 2d, e), suggesting that caRNAs are even more enriched in SEs than in enhancers.

To test whether the detected enrichment is sensitive to the precise boundaries of SEs, we extended the boundary of the SE to the boundary of the overlapping gene whenever a SE is fully embedded in a gene. This boundary extension resulted in 875 HUVEC SEs, covering 3.1% (94,493,925 bp) of the genome. These extended SEs accounted for 15.1% of the RNA ends of iMARGI read pairs, representing 4.9-fold increase of odds than genome average (odds ratio = 5.6, p value < 2.2e−16, d.f. = 1, chi-square test). The enrichment in the extended SEs (odds ratio = 5.6) remained greater than the enrichment in all enhancers (odds ratio = 2.7), suggesting that the enrichment of caRNAs in SEs is not sensitive to the precise boundaries of SEs. Taken together, iMARGI-identified caRNAs were enriched with eRNAs and even more enriched with transcripts from SEs, i.e., seRNAs. From here on, our analysis was based on the SEs with extended boundaries. We call a SE as geneSE, where gene is the gene (or one of the genes) overlapping with this SE. We call SE-derived caRNAs as se-caRNAs.

**Emergence of a RNA–chromatin interaction network among SEs.** More than 17% of the iMARGI read pairs have either the RNA or DNA end mapped to SEs. The SEs were 1.7-fold more likely to be mapped by the RNA ends than the DNA ends (p value = 0.05, t test), suggesting that SEs contribute to caRNAs more often than being the genomic targets of caRNAs. More iMARGI read pairs mapped to SEs as compared to genome average (odds ratio = 6.5, p value < 2.2e−16, d.f. = 1, chi-square test). This enrichment suggests a subnetwork composed of SEs in the RNA–chromatin interactome[26].

To characterize this subnetwork, we counted the number of iMARGI read pairs between any two SEs and normalized these counts by the total number of uniquely mapped read pairs in each sample. A pair of SEs was called interacting when their normalized counts were above the 95th percentile of all the normalized counts at Day 0. This analysis resulted in 1787, 2777, and 3785 interacting SE pairs at Days 0, 3, and 7, respectively (Supplementary Fig. 4d, e). Among these identified SE interactions, the number of interchromosomal SE pairs increased from 506 (Day 0) to 2139 (Day 3) and subsequently to 3253 (Day 7; Supplementary Fig. 4b, d). In comparison, the number of intrachromosomal interactions did not increase (Supplementary Fig. 4b, e). These data suggest that some changes of gene expression in dysfunctional ECs are caused by interchromosomal RNA–DNA interactions between SEs, i.e., an interchromosomal RNA activation hypothesis.

**Hubs of interchromosomal RNA–DNA interaction networks.** To provide further clues to test this hypothesis, we aimed to

identify a small portion of the SE interactions likely more important for EC dysfunction. The degree distribution of these SE networks followed the power law (Supplementary Fig. 4c). Thus, these SE networks are hierarchical networks with a small number of highly connected central nodes (a.k.a. hubs)[27,28]. We identified the hubs as those SEs with degrees 60 or greater (≥95th percentile of all the degrees). This analysis resulted in 1, 14, and 25 hubs involved in 130, 1652, and 2514 interchromosomal connections at Days 0, 3, and 7, respectively (Fig. 3a).

The only hub SE on Day 0 overlapped with MALAT1. Following the rule defined above, we call this Malat1SE. The large number of caRNAs transcribed from Malat1SE is expected because the MALAT1 lncRNA interacts with a large amount of transcription active genomic regions[29]. The number of hubs increased from 1 (Day 0) to 14 (Day 3) and to 25 (Day 7; Fig. 3a). Furthermore, every hub of the preceding time point appeared as a hub in the following time point. Such continuity is reproduced in biological replicates (Supplementary Fig. 4f). These data suggest a continuous expansion of hubs over the progression of EC dysfunction.

**H + T-induced hub SEs contain driver genes of EC dysfunction.** We next asked whether the emergent hubs in the H + T treatment time course contribute to EC dysfunction. The H + T-induced hub SEs included Serpine1SE, Fndc3bSE, Thbs1SE, Pvt1SE, Smad3SE, Runx1SE, and VwfSE (Supplementary Table 3). The genes embedded in these hub SEs encode key activators of inflammation and thrombosis, including SERPINE1, THBS1, and VWF, all of which have also been shown to be elevated in diabetes[30–34], inhibitors of angiogenesis, including THBS1[31] and RUNX1[35], drivers of EndoMT including SMAD3[20], and several others promoting EC dysfunction including TRIO[36], EXT1[37], and PDE4D[38] (Fig. 3a). Thus, the genes embedded in the H + T-induced hub SEs are critical to the transition from healthy to dysfunctional ECs, constituting a core feature of the interchromosomal RNA–chromatin interactome.

**Inhibition of selected caRNA suppresses EC dysfunction.** The aforementioned data, together with our previous study[5], suggest a model in which interchromosomal RNA–chromatin interactions activate critical genes in the target genomic regions to contribute to EC dysfunction. To test this thesis, we perturbed the RNA–chromatin interaction between two SEs by inhibiting the ncRNA transcribed from a SE. Among all the H + T-induced interacting SEs, 26 SEs do not contain any coding gene and contain at least one lncRNA gene. Among the lncRNAs contained in these SEs, only LINC00607 was reported as a H + T-induced lincRNA by scRNA-seq analysis with a threshold of FDR < 1e−32 (Fig. 1d, e).

The SE containing LINC00607 is Linc607SE (Supplementary Fig. 6), which did not interact with the hub SE (Malat1SE) in control ECs (Day 0), but exhibited H + T-induced interactions with several hub SEs, including Serpine1SE (Fig. 3a, b and Supplementary Fig. 5). Serpine1SE contains the SERPINE1 gene, which encodes PAI-1, a key regulator of EC dysfunction linking hyperglycemia, inflammation, and EndoMT[16,22]. SERPINE1 is also one of the top H + T-induced genes in ECs from scRNA-seq (Fig. 1d, e). Notably, Hi-C data did not reveal interactions between Linc607SE and Serpine1SE in ECs from any time point (Fig. 3c).

To test whether the transcripts derived from Linc607SE have an effect on H + T-induced SERPINE1 expression, we designed two locked nucleic acid (LNA) GapmeRs targeting LINC00607 (Fig. 4a). Both LNAs significantly reduced the RNA levels of LINC00607 in H + T-treated HUVECs (Fig. 4b). SERPINE1

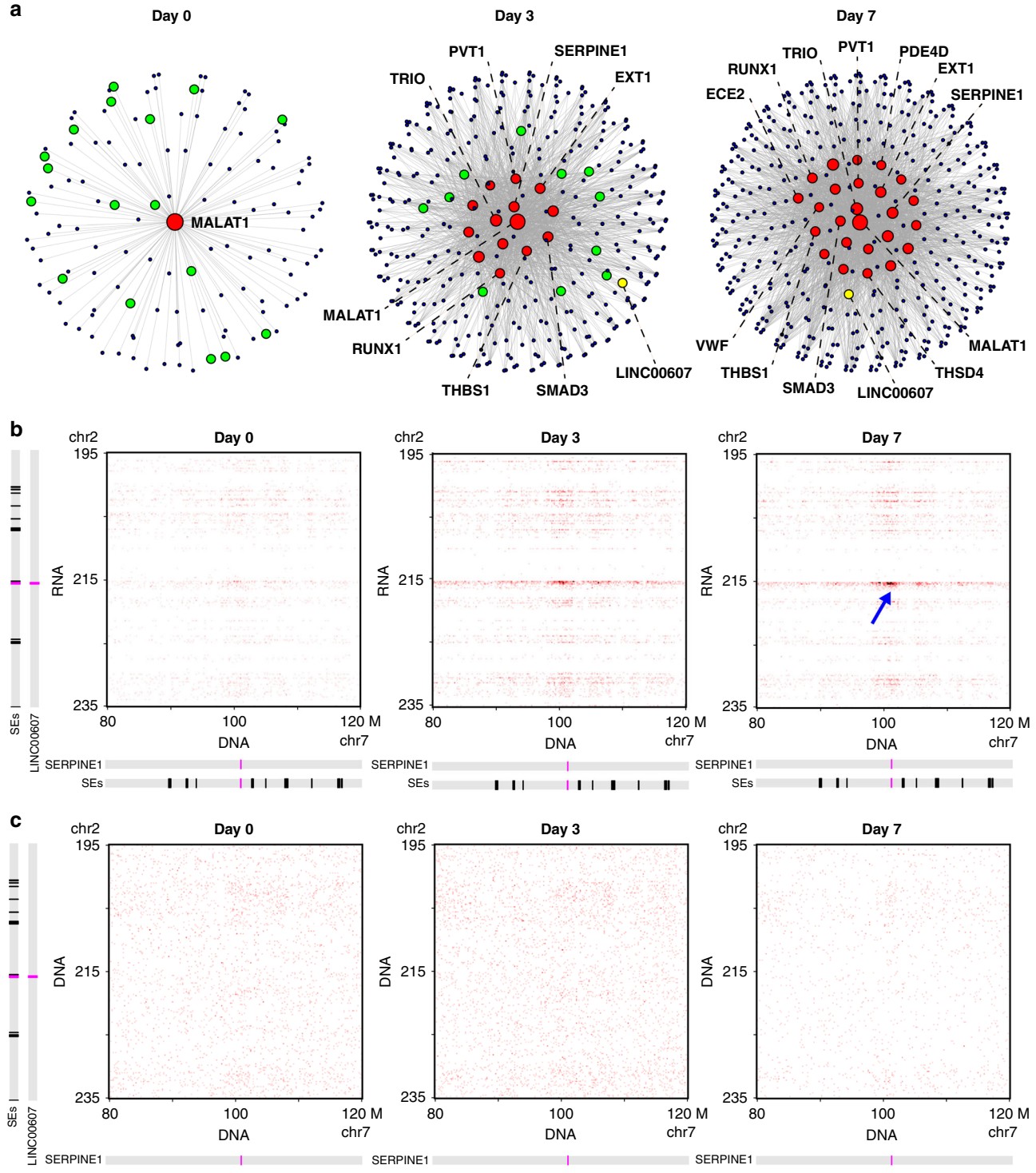

**Fig. 3 Hub SEs induced by H + T contain driver genes of EC dysfunction. a** Interchromosomal RNA–chromatin interaction networks. Each node is a SE. Each edge is an interchromosomal interaction between two SEs. Red nodes: hub SEs. Green nodes: SEs that became hub SEs at the next time point. Yellow node: *Linc607SE*. The SEs that contain EC dysfunction-related genes are marked with the gene name. **b, c** Contact matrices between chromosome 2 (195–235 Mb) and chromosome 7 (80–120 Mb). **b** iMARGI derived contact matrices from the RNA (rows) to DNA (columns) in Days 0 (left), 3 (middle), and 7 (right) ECs. Blue arrow: interactions between *Linc607SE* and *Serpine1SE* established at Day 7. **c** Hi-C derived contact matrices. Locations of SEs, *LINC00607*, and *SERPINE1* are marked on the left and at the bottom. Resolution = 200 kb.

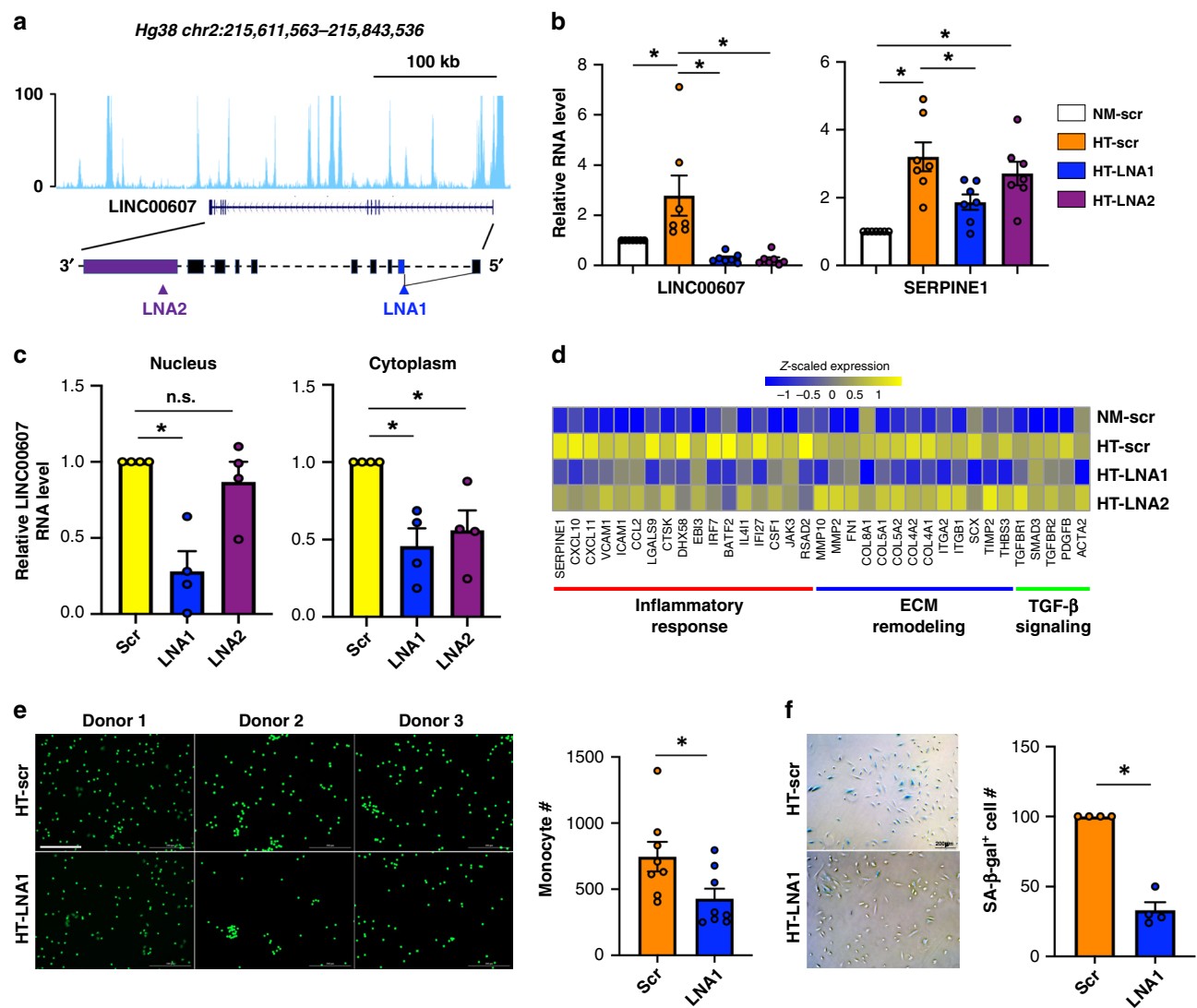

**Fig. 4 Inhibition of *LINC00607* attenuates SERPINE1 induction and EC dysfunction. a** Illustration of *LINC00607* genomic locus and gene structure and LNA GapmeRs targeting of LINC00607 RNA. **b** HUVECs were transfected with scramble (scr) or two LNAs targeting LINC00607. LINC00607 RNA and SERPINE1 mRNA levels were quantified. Data represent mean ± SEM from seven independent experiments. **c** qPCR of LINC00607 in subcellular fractionations of HUVECs transfected with scr, LNA1, or LNA2. Data represent mean ± SEM from four independent experiments. **d** RNA-seq was performed with cells transfected as in **b** in biological replicates. Heatmap is plotted based on *z*-scaled log-transformed gene expression levels. **e**, **f** ECs transfected with scr or LNA1 and then treated by H + T were used in **e** monocyte adhesion assay and in **f** SA-β-gal assay. In **e**, the number of attached monocytes to ECs were quantified. Data represent mean ± SEM from eight independent experiments performed with peripheral blood-derived monocytes from four individual donors. Representative images show the attachment of fluorescently labeled peripheral blood-derived monocytes to ECs for experiments performed with monocytes from three different donors. In **f**, ECs with positive β-gal staining were quantified. The positively stained cell number in scr control was set to 100 (%). **f** Data represent mean ± SEM from four independent experiments respectively. Scale bar = 200 μm. * denotes $p = 0.0289$, 0.0013, 0.001, 0.0001, 0.0025, and 0.024 from left to right (in **b**) and $p = 0.0023$, 0.0079, and 0.0249 from left to right (in **c**) between indicated groups based on ANOVA followed by Bonferroni post hoc test (in **b** and **c**); $p = 0.0039$ (in **e**) and 0.0014 (in **f**), as compared to scr group based on two-tailed paired *t* test (in **e** and **f**). Source data are provided as a Source data file.

expression was also reduced by both LNA1 and LNA2, although only the reduction by LNA1 reached the statistical significance of $P < 0.05$ (Fig. 4b).

To investigate why LNA2 did not affect the *SERPINE1* expression as efficiently as LNA1, we queried whether the two LNAs exerted differential effects on LINC00607 RNA depending on its subcellular localization. Based on the ENCODE[39] data, among all the predicted *LINC00607* transcripts, LINC00607:3 is the most abundant and nucleus-enriched transcript in HUVECs, which we confirmed in ECs (Supplementary Fig. 6). It is also the only validated transcript on NCBI (reference sequence: NR_037195.1), which the LNAs were designed to target. Using

subcellular fractionation, we found that LNA2 did not inhibit the LINC00607 RNA level in the nucleus as efficiently as LNA1, although both LNAs inhibited the levels of cytoplasmic LINC00607, (Fig. 4c and Supplementary Fig. 6). These data support that nuclear-localized LINC00607 RNA promotes SERPINE1 transcription in dysfunctional ECs.

We also performed RNA-seq to obtain additional information on the effect of LINC00607 knockdown. Consistently, RNA-seq confirmed that H + T-induced *SERPINE1* mRNA level was substantially inhibited by LNA1, but not as effectively by LNA2. Moreover, genes known to be regulated by *SERPINE1*, e.g., *FN1* (refs. [40,41]) and *COL4* (refs. [40,42]) were also suppressed

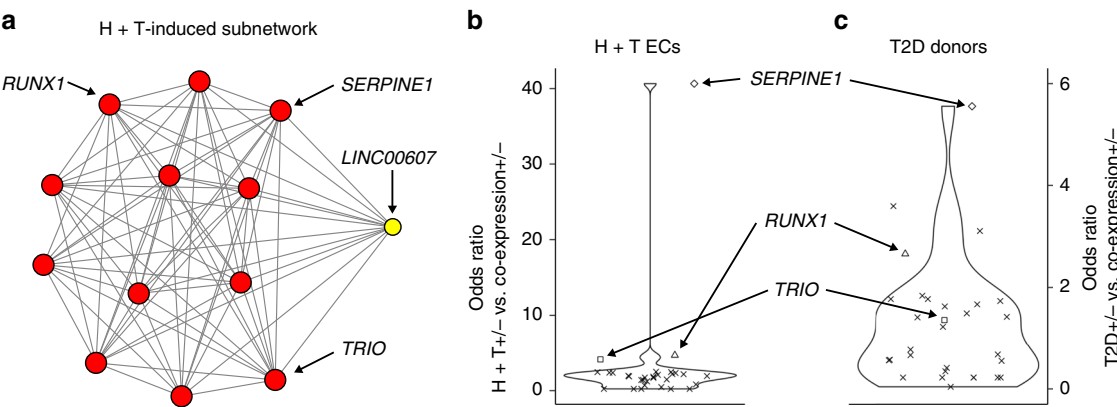

**Fig. 5 Increased single-cell *LINC00607–SERPINE1* co-expression in dysfunctional ECs. a** H + T Day 7 induced interactions (edges) between *Linc607SE* (yellow node) and hub SEs (red nodes). **b, c** Odds ratio (*y*-axis) between single-cell co-expression levels and the health status of ECs, including H + T vs. control ECs (**b**) and diabetic (T2D) vs. healthy vascular endothelium (**c**). Each dot corresponds to a gene in any *Linc607SE*-interating SE (**a**). A large odds ratio corresponds to a positive association between dysfunction (H + T Day 7 for **b** and diabetic for **c**) and the single-cell co-expression of this gene with *LINC00607*. Source data are provided as a Source data file.

by LNA1 (Fig. 4d). To link the molecular changes to EC function, we assessed monocyte adhesion to ECs and senescence-associated β-gal staining of ECs, two functional readouts of *SERPINE1* induction[43,44]. In line with the gene expression change, LNA1 suppressed donor-derived monocyte adhesion to HUVECs and the EC senescence marker (Fig. 4e, f). Collectively, these data support that perturbing the RNA–chromatin contacts, exemplified by LINC00607 knockdown, led to the suppression of SERPINE1 and endothelial dysfunction.

**Co-expression of *LINC00607* and *SERPINE1* in dysfunctional ECs.** To explore an alternative approach for testing the regulatory link between *LINC00607* and *SERPINE1*, we reasoned that if LINC00607 RNA promotes transcription of *SERPINE1* in dysfunctional ECs, we shall expect *LINC00607* and *SERPINE1* co-expressed in the same single cells more often in the dysfunctional ECs than in control ECs. To this end, we compared the number of single cells expressing both *LINC00607* and *SERPINE1* (co-expression+) with the number of single cells expressing only *LINC00607* (co-expression−) between the H + T-treated and control ECs (diamond dot, Fig. 5b). The H + T-treated ECs exhibited a 9.1-fold increase in the odds of co-expression than the control ECs (odds ratio = 40.7, *p* value < 2.2e−16, d.f. = 1, chi-square test), suggesting that H + T-induced *SERPINE1*-expressing cells tended to coincide with those single cells that expressed *LINC00607*.

Next, we asked whether *Linc607SE*-interacting SEs contained any other genes that also exhibited H + T-induced co-expression with *LINC00607* in the same single cells. We identified all the 30 genes contained in *Linc607SE*-interacting SEs at H + T Day 7 (Fig. 5a) and repeated the association test by replacing *SERPINE1* with every other gene (dots, Fig. 5b). While *SERPINE1* exhibited the largest degree of H + T-induced single-cell co-expression with *LINC00607* (diamond dot, Fig. 5b), 21 genes including *RUNX1* and *TRIO* also exhibited increased co-expression with *LINC00607* in H + T-treated ECs, as compared to control ECs (odds ratio > 1, Fig. 5b). These data suggest that the SEs exhibiting H + T-induced interactions are more likely to co-express their harbored genes in the same single cells in dysfunctional ECs than in healthy ECs.

**Diabetes-associated co-expression in single cells.** Next, we asked if vascular endothelium from diabetic donors also exhibits increased co-expression of *LINC00607* and *SERPINE1* in the same single cells, as compared to vascular endothelium from healthy control donors. To this end, we isolated ECs for scRNA-seq from mesenteric arteries from two healthy and two type 2 diabetic (T2D) donors (Supplementary Table 4). One of the T2D donors had over 10 years of diabetic history and the other had untreated T2D, both of which are expected to have impaired EC function.

The diabetic donors' ECs exhibited a 3.8-fold increase in the odds of co-expressing *LINC00607* and *SERPINE1* than the ECs from healthy donors (odds ratio = 5.6, *p* value = 5e−7, d.f. = 1, chi-square test), suggesting a corresponding increase of single-cell co-expression in diabetes. Compared to the other 29 genes embedded in the 12 *Linc607SE*-interacting SEs, again *SERPINE1* ranked first by odds ratio in both cultured ECs and donor-derived ECs (diamond dots, Fig. 5b, c), revealing another level of consistence of diabetes-associated single-cell transcriptomes with that of our in vitro model. Taken together, diabetic conditions induced consistent single-cell co-expression changes of *LINC00607* and *SERPINE1* in human donor-derived ECs as in our in vitro model. Nevertheless, compared to data from cultured ECs, in which *RUNX1* ranked second by odds ratio, two other genes in iMARGI-identified *Linc607SE*-interacting SEs exhibited stronger single-cell co-expression with *LINC00607* than *RUNX1* in data from donors' ECs (triangle dots, Fig. 5b, c). These data reveal that our in vitro disease model does not completely recapitulate the molecular changes in T2D patients.

## Discussion
Global changes of RNA–chromatin interaction patterns suggest a global force of nuclear reorganization. Although this global force does not exclude any gene- or sequence-specific mechanisms that specifically tether an RNA to a target genomic sequence, gene- or sequence-specific tethering alone is unlikely to accumulate into the observed global changes.

A conceivable model is a global relaxation of the globular structure of each chromosome, obscuring the boundaries of chromosome territories, thus making the transcripts from one chromosome more likely to diffuse to the spatial proximity of some genomic regions of another chromosome. This model can be checked by three methods. First, assumption: there are two competing physical means to organize the chromatin polymer into a globular structure, specifically fractal globule that does not form local knots and equilibrium globule with local knots that

does not allow for a simultaneous global relaxation[45]. This global relaxation model assumes that the chromosomes are folded as fractal globule instead of as equilibrium globule. This assumption is supported by independent confirmations of a reverse linear relationship between the genomic interaction frequencies and the genomic distances[45–48].

Second, if this global relaxation model were correct, we would expect a decrease in the intrachromosomal DNA contact frequency at any given genomic distance in dysfunctional ECs. Indeed, Hi-C data at Days 3 and 7 exhibited progressive decrease of DNA contact frequency at any given genomic distance (Supplementary Fig. 7a). Furthermore, we would expect to see that TAD structures are weakened in dysfunctional ECs. Indeed, H + T induced progressive decrease of proportions of reads mapped within TADs (Supplementary Fig. 7b).

Third, if the increasing obscurity of chromosomal territories allowed diffusion-based RNA access to a SE on another chromosome, we would expect the spatial distance between RNA-producing genes and the target SEs to decrease. Indeed, the distances between Linc607SE and Serpine1SE are shortened in H + T-treated ECs based on DNA FISH assay (Supplementary Fig. 8). Finally, because this distance change is mutual, we would expect that the increase of RNA–chromatin interactions is bidirectional. Indeed, both increasing amounts of LINC00607 RNA attached to Serpine1SE and increasing amounts of Serpine1SE transcripts attached to chromosome 2 near LINC00607 were detected (Fig. 2c and Supplementary Fig. 5), suggesting that the emergent RNA–chromatin interactions between the two SEs were reciprocal.

In the context of HG and TNFα-induced EC dysfunction, we observed abundant interchromosomal RNA–DNA interactions enriched among SEs that progressively increased as the treatment prolonged (Fig. 2). Specifically, a number of EC dysfunction driver genes (e.g., SERPINE1, THBS1, and VWF) are embedded in the emergent hub SEs for these interactions (Fig. 3). These data are in line with the previous study, emphasizing the central role of inflammatory SEs in transcriptional induction during EC activation, the initial stage of EC dysfunction[14]. Compared to the use of a much shorter time frame (1–4 h) and a high dosage of TNFα in the previous study, our findings provided insights into sustained transcriptional activation involving SE-derived caRNAs that contribute to chronic EC dysfunction in disease states, such as diabetes.

Using one of the emergent SE-derived interacting pairs (i.e., Linc607SE-Serpine1SE) as an example, we showed that perturbation of a caRNA, putatively by disrupting the RNA–chromatin interaction, suppressed the H + T-induced SERPINE1 expression (Fig. 4). Such mode of action is unlikely to be a unique case for one particular gene, as inhibition of LINC00607 also leads to suppression of other genes embedded in the SEs that exhibited increased interactions with LINC00607 under H + T treatment, including TRIO, COL4, and THSD4 (Supplementary Table 5). In line with this data, the correlation analysis using scRNA-seq data from H + T-treated ECs and diabetic donor-derived ECs also showed an increase in the odds of co-expressing LINC00607 and these genes in the same single cells (Fig. 5). In addition, the effect of caRNAs most likely requires participation of other transcriptional activators (e.g., transcriptional factors and co-activators)[2] as overexpression of LINC00607 RNA alone did not result in a significant induction of SERPINE1 (Supplementary Fig. 9). Likewise, the proposed interchromosomal RNA–DNA interaction mechanisms are unlikely to involve one specific lncRNA as there are other lncRNAs embedded in the emergent hub SEs that form increased RNA–chromatin interactions under H + T treatment (Supplementary Table 3). Compared to our previous study suggesting the role of one enhancer-derived lncRNA in transcriptional induction through an interchromosomal RNA–DNA

interaction[5], our current study revealed genome-wide prevalence of these RNA–chromatin contacts and provided evidence to support their functional importance in EC biology. Future studies are warranted to further dissect the molecular mechanisms involving caRNAs, and delineate their functional relevance in health and disease.

The single-cell transcriptome profiles in ECs clearly demonstrate a time-dependent activation of pro-inflammatory response, ECM remodeling, and pro-fibrotic TGF-β signaling (Fig. 1). These disease-driving cell state transitions are a common theme in many diseases, including those with EC dysfunction at play and beyond[49–51]. However, the mechanisms linking inflammation and tissue fibrosis are not well understood. Among the genes embedded in H + T-induced hub SEs, the induction of SERPINE1, THBS1, and VWF typically promotes inflammation, thrombosis, and ECM remodeling[30–34], while SMAD3 is a canonical activator for TGF-β signaling and fibrosis[20] (Fig. 3). The strong and sustained activation of these hub SEs for interchromosomal RNA–DNA interactions may provide a mechanistic link at the 3D genome organizational level to connect these prominent cellular pathways driving various diseases. It would be interesting to explore in future studies whether the observed emergence of SE network can be recapitulated in other cell types and in other disease contexts.

Notably, HG can induce a metabolic memory, i.e., persistent changes detectable even after the switch to normoglycemia[52–54]. Interestingly, although replenishing ECs with control media after 3 days of H + T treatment partially reversed the changes in cell morphology (Supplementary Fig. 10a), and eNOS and ICAM1 mRNA levels, those of LINC00607 and SERPINE1 could not be reversed (Supplementary Fig. 10b). These data suggest that the mechanisms involving the RNA–chromatin contacts may be an important, yet poorly elucidated, layer underlying the undesirable metabolic memory associated with diabetic complications.

In conclusion, by using a systems biology approach, our study provides an example of how RNA–chromatin contacts can affect gene expression and cellular states, as exemplified by endothelial dysfunction, a dynamic and vital process underlying a variety of disease.

## Methods

**Cell lines.** HUVECs (passages 5–8) from pooled donors were used in this study. The cells have been tested negative for mycoplasma contamination and pre-screened to demonstrate stimulation-dependent angiogenesis and key EC signaling pathways. ECs were cultured at 37 °C with 5% $CO_2$ in M199 (Sigma M2520) supplemented with 20% FBS (Hyclone, SH30910.02), β-EC growth factor (Sigma, E1388), and 100 units/mL penicillin and 100 mg/mL streptomycin (Thermo Fisher Scientific).

**Cell transfection and stimuli.** For LINC00607 knockdown experiments, two antisense LNA GapmeRs specifically targeting two different regions of LINC00607 (NR_037195.1) were designed and purchased from QIAGEN (Supplementary Table 6). LNAs were separately transfected into HUVECs with Lipofectamine RNAiMAX following the protocol provided by the manufacturer. The cells were cultured in M199 complete medium 4–6 h after transfection, and then subjected to HG and TNFα treatment. HG condition was generated by adding D-glucose into the culture media to a final concentration of 25 mM. TNFα was added to the culture media to a final concentration of 5 ng/mL. A total of 25 mM D-mannitol was used as an osmolarity control. Typical EndoMT stimuli TGF-β and IL-1β were added to the culture media at a final concentration of 10 and 1 ng/mL, respectively.

For LINC00607 overexpression, the cDNA of LINC00607:3 isoform was amplified by using SMARTer® RACE 5′/3′ Kit (Takara Bio USA) from HUVEC total RNAs, and cloned into a pcDNA3.1(+) vector. The sequence was confirmed by Sanger sequencing by alignment to NR_037195.1. Plasmid transfection was performed using the Cytofect™ HUVEC Transfection kit (Cell Applications) following the manufacturer's protocol in 6-well or 12-well plates. Cells were harvested at 48 h post transfection.

**RNA extraction and quantitative PCR.** Total RNA was isolated using TRIzol reagent. cDNAs were synthesized using PrimeScript™ RT Master Mix containing

both Oligo-dT primer and random hexamer primers. qPCR was performed with Bio-Rad SYBR Green Supermix following the manufacturer's suggested protocol, using the Bio-Rad CFX Connect Real Time system. All primer sequences used in qPCR amplification are listed in Supplementary Table 6.

**(Immuno)fluorescent staining**. HUVEC cells were plated on coverslips (pre-coated with poly-L-lysine and 0.1 M collagen or 50 µg/mL fibronectin solution), and treated with HG and TNF-α, or cultured in control conditions. For VE-cad and α-SMA immunostaining, cells were washed with PBS and fixed with 100% methanol for 15 min in −20 °C. The fixed cells were rinsed three times with PBS for 5 min each and incubated in a blocking buffer containing 5% BSA in PBST (PBS with 0.1% Triton X-100) for 1 h. Mouse anti-human α-SMA antibody (Abcam, ab124964, at 1:500 dilution) and rabbit anti-VE-cad antibody (Abcam, ab33168, at 1:200 dilution) were added to cells in blocking buffer, and incubated overnight at 4 °C. From this step on, cells were protected from light. After rinsing in PBST three times (10 min each), cells were incubated with a cocktail of Alexa Fluor 488-conjugated goat anti-mouse (for α-SMA staining, Fisher, A10680, 1:500 dilution) and Alexa Fluor 555-conjugated donkey anti-rabbit (for VE-cad staining, Fisher, A31572, 1:500 dilution) antibody in blocking buffer at room temperature (RT) for 1 h. Samples were washed three times in PBST for 10 min each, then stained with DRAQ5 (Abcam, AB108410, 1:1000 dilution) in PBST at RT for 15 min. The fluorescence images were taken with an Echo revolve fluorescence microscope.

**scRNA-seq of HUVECs and data analysis**. Mannitol control (Day 0), HG + TNF (Day 3), and HG + TNF (Day 7) with biological duplicates were prepared as single-cell samples for sequencing using Drop-seq protocol with 10× Genomics Chromium 3′ expression protocol. There are >60 M reads/sample, 4000–15000 cells/sample. scRNA-seq data have been processed using the standardized pipeline provided by 10× Genomics (v3.0) and aligned to human hg38 reference transcriptome. The R package Seurat (v2.3.4) was used to analyze scRNA-seq data following published guidelines[55]. First, we performed a filtering step using well-established quality control metrics. Rare cells with very high numbers of genes (potentially multiplets), as well as high mitochondrial percentages (low-quality or dying cells often present mitochondrial contamination) were removed. We set the upper threshold for both of those features as the 99th percentile of their distribution in each sample. In addition, cells exhibiting a gene count <300 were filtered out as potential low-quality cells or empty droplets. Filtration led to the removal of ~2% of the total cells (from 60,841 to 59,605 total cells across all the samples).

Next, data were normalized by default in Seurat. We employed the global-scaling LogNormalize method, which normalizes the gene expression measurements for each cell by the cell total expression, multiplies it by a scale factor (10,000 by default), and log-transforms as $\log(x + 1)$ the result. We selected highly variable genes (HVGs) and scaled the gene expression data for downstream analysis. HVGs were calculated as default in Seurat by using the log(variance to mean ratio) (logVMR) for each gene in the dataset. Genes were then sorted by decreasing logVMR, from which we extracted the top 1000 HVG. Unwanted sources of variation, such as mitochondrial expression and number of detected molecules per cell, were regressed out and the expression of each HVG was scaled to obtain a $z$-score for each gene across all the single cells in the dataset. PCA was performed across all cells and the top 1000 HVGs using the scaled $z$-scored expression values. The first 20 significant PCs were then used as input to the $t$-SNE algorithm. $t$-SNE plots were used also to show the expression level of selected genes in each single cell across the time course. The RNA levels are represented by log-normalized unique molecular identifier (UMI) counts. Values below the 10th percentile and above the 90th percentile were clipped using the appropriate parameters in FeaturePlot.

Differential expression analysis was performed only using the subpopulation of ECs extracted as described above. We used the nonparametric Wilcoxon test (default in Seurat and one of those that globally perform the best according to Soneson et al.)[56]. The test was performed using default parameters in Seurat. Thus, only genes expressed in at least 10% of the cells in a sample were used in the analysis. To extract differentially expressed (DE) genes, the threshold for the log fold-change of the average expression (logFC_avg) between two samples was left to 0.25 (either for upregulated or downregulated genes). We used a pseudocount of 1 (as default) to be added to the averaged expression values when calculating logFC. This prevents extremely lowly expressed genes from dominating the DE analysis. The top DE genes in single ECs were plotted on a $z$-scaled binned expression heatmap. In order to generate the heatmap, we started from the $z$-scaled data and ordered all the cells by increasing SERPINE1 expression (per each sample separately). Then, cells were binned per 100 cells for the analysis: 269 bins in Day 0, 177 bins in Day 3, and 148 bins in Day 7. Before plotting, we clipped values above 2.5 and below −2.5 as default in Seurat DoHeatmap. Pathway enrichment analysis for DE genes was performed using the Database for Annotation, Visualization and Integrated Discovery (DAVID, https://david.ncifcrf.gov/). DAVID uses a more conservative modified one-sided Fisher Exact test for extracting enriched pathways (number of genes in pathway minus 1 in the contingency table). Thus, a modified Fisher Exact $p$ value (named EASE score) is used to detect the significant enrichment (EASE score < 0.1 as default).

**scRNA-seq of donor-derived arterial ECs and data analysis**. Human tissue studies were conducted on deidentified specimens obtained from the Southern California Islet Cell Resource Center at City of Hope. The research consents for the use of postmortem human tissues were obtained from the donors' next of kin and ethical approval for this study was granted by the Institutional Review Board of City of Hope (IRB #01046). All work presented was performed in compliance with relevant ethical regulations. T2D was identified based on diagnosis in the donors' medical records, as well as the measurement of 6.5% or higher of glycated hemoglobin A1c.

Single-cell RNA-seq was performed in human mesenteric arterial ECs in two healthy and two T2D donors, following the procedures we recently described[57]. Briefly, the arterial intima was gently dissociated from the arterial wall using a scalpel. The single-cell suspension and scRNA-seq libraries were prepared as described above. There are >126 M reads/sample and 1800–4400 cells/sample. scRNA-seq data were processed using the standardized pipeline provided by 10× Genomics (v3.0) and aligned to human hg38 reference transcriptome. The R package Seurat (v2.3.4) was used to analyze scRNA-seq data following published guidelines[55]. First, we performed a filtering step using well-established quality control metrics. Rare cells with a very high number of genes (potentially multiplets), as well as high mitochondrial percentages (low-quality or dying cells often present mitochondrial contamination) were removed. We set the upper threshold for the number of genes as the 98th percentile of its distribution in each sample, while the maximum mitochondrial percentage per cell was set at 20%. In addition, cells exhibiting a gene count lower than 300 were filtered out as potential low-quality cells or empty droplets. Filtration led to the removal of ~12% of the total cells (from 12,815 to 11,243 total cells across all the samples).

Next, data were normalized by default in Seurat following the same steps performed for the HUVEC data: (1) global-scaling normalization using the LogNormalize method, (2) selection of HVGs and scaling of the gene expression data, also removing unwanted sources of variation, such as mitochondrial expression and number of detected molecules per cell, and (3) performing PCA across all cells and the top 1000 HVGs using the scaled $z$-scored values. The first 20 significant PCs were then used as input to the $t$-SNE algorithm. In order to select ECs for analysis, we performed shared nearest neighbor (SNN) clustering and visually selected clusters showing high expression levels of CDH5, which encodes VE-cad (Supplementary Fig. 11). SNN clustering was performed using a resolution of 0.4. In Supplementary Fig. 11c, the RNA levels of CDH5 are represented by log-normalized UMI counts. Values below the 10th percentile and above the 90th percentile were clipped using the appropriate parameters in FeaturePlot.

Differential expression analysis was performed only using the subpopulation of ECs extracted as described above. We used the nonparametric Wilcoxon test (default in Seurat and one of those that globally perform the best according to Soneson et al.)[56]. The test was performed using default parameters in Seurat. Thus, only genes expressed in at least 10% of the cells in a sample were used in the analysis. To extract differentially expressed (DE) genes, the threshold for the log fold-change of the average expression (logFC_avg) between two samples was left to 0.25 (either for upregulated or downregulated genes). We used a pseudocount of 1 (as default) to be added to the averaged expression values when calculating logFC. This prevents extremely lowly expressed genes from dominating the DE analysis.

**Hi-C**. Hi-C was performed using an Arima-HiC kit (Arima Genomics, Inc.) following the manufacturer's manual. Hi-C data were processed and plotted using HiCtool (v2.2)[58]. Data were normalized with the matrix balancing approach performed by Hi-Corrector[59] and incorporated into HiCtool. TAD and compartment analyses were performed as well using HiCtool. MoC among TAD boundaries was calculated following the approach, as described by Zufferey et al.[23] (Supplementary Fig. 3c).

To avoid any bias given by the different sequencing depth among samples, we randomly sampled 190 million read pairs from every Hi-C sequencing library before performing the following analyses. Pearson correlation matrix was derived from the observed/expected contact matrix at 1 Mb resolution, and the eigenvector corresponding to the first principal component of the Pearson correlation matrix was used to identify A/B compartments. We used normalized data binned at 40 kb to calculate the average interaction frequency at each genomic distance between 40 kb and 20 Mb. The curves (one per sample) show this function genome-wide, meaning the average interaction frequency at each distance averaged across all chromosomes (Supplementary Fig. 7a). Supplementary Fig. 7b shows the distributions of the proportions of reads mapped within TADs for all the chromosomes. The proportion of reads mapped within a TAD in a sample was calculated on the normalized data at 40 kb by summing the number of interactions within that TAD and dividing it by the total number of uniquely mapped read pairs in that sample. TAD boundaries were calculated from normalized contact data at 40 kb resolution as well.

**iMARGI assay and data analysis**. iMARGI was performed as described in our recent report[7]. Briefly, iMARGI started with crosslinking cells using 1% formaldehyde, collecting nuclei, followed by fragmenting RNA and DNA in nuclei using RNase I and restriction enzyme AluI. A specifically designed linker sequence was introduced to the permeated nuclei to ligate with the fragmented RNA and

subsequently ligate with spatially proximal DNA. After the ligation steps, nuclei were lysed and crosslinks were reversed. Nucleic acids were purified and subsequently treated with exonucleases to remove any linker sequences that were not successfully ligated with both RNA and DNA. The desired ligation products in the form of RNA–linker–DNA were pulled down with streptavidin beads. The RNA part of the pulled down sequence was reverse transcribed into cDNA, resulting in a complementary strand of (5′) DNA–linker–cDNA (3′). Single-stranded DNA–linker–cDNA was then released from streptavidin beads, circularized and re-linearized, producing single-stranded DNA in the form of left.half. Linker–cDNA–DNA–right.half.Linker. The two halves of the linker (left.half.Linker and right.half.Linker) served as templates for PCR amplification. The linearized DNA was amplified with NEBNext PCR primers for Illumina, size-selected, and subjected to 100 cycles of pair-end sequencing with an Illumina Hi-seq 4000. Approximately 300 million read pairs were obtained per sample. Sequencing data were aligned to the hg38 reference genome using STAR (v2.5.4b). In-house scripts were used to deduplicate (FastUniq v1.1) and parse (Samtools v1.6) the mapped read pairs to obtain the BEDPE file with the uniquely mapped read pairs.

SEs for HUVECs were downloaded from dbSUPER[25] and their genomic coordinates were converted from hg19 to hg38 using the UCSC Lift Genome Annotations tool (https://genome.ucsc.edu/cgi-bin/hgLiftOver). The 912 SEs were classified into three categories: not overlapping any gene (84), overlapping one or more genes, but not fully embedded within any of them (449), and overlapping one or more genes and fully embedded within at least one of them (379). For these 379 SEs, we extended the start and end coordinate per each SE $SE_i$ as following: (1) we extracted all the genes $SE_i\_genes$ embedding $SE_i$; and (2) we updated $SE_i$ start coordinate with the minimum start coordinate of $SE_i\_genes$, and $SE_i$ end coordinate with the maximum end coordinate of $SE_i\_genes$. In the end, we obtained 875 SEs with extended boundaries that were used in the following analysis. The sum of the lengths of these SEs is 94,493,925 bp with 875 SEs in total. If we consider the entire genome length (3,088,286,401 bp), 3.1% of the entire genome is occupied by SEs.

To characterize SE networks, we counted the number of iMARGI read pairs between any two SEs and normalized these counts by the total number of uniquely mapped read pairs in this sample. A pair of SEs was called interacting when their normalized counts were above the 95th percentile of all the normalized counts at Day 0 (i.e., 2e−7 for replicate 1, and 2.5e−7 for replicate 2). SE hubs were identified as those SEs with degrees 60 or greater (≥95th percentile of all the degrees). SE networks (Fig. 3a and Supplementary Fig. 4d, e) were plotted using igraph, a custom function based on the R package (v1.2.4.1). The network in Fig. 5a was plotted using Cytoscape (v3.5.1). The coverage plot (Supplementary Fig. 5a) was generated using the R package karyoploteR (v1.10.5), while the iMARGI read pair plot (Fig. 2c) was created using the R package "Gviz" (v1.28.3). Statistical analyses were performed in R.

**RNA-seq and data analysis.** HUVECs were harvested for RNA extraction using Trizol reagent. Total RNA (200 or 500 ng) per sample was subjected to library construction using KAPA mRNA HyperPrep Kit (Roche Diagnostics) following the manufacturer's manual. The libraries were sequenced in HiSeq2500 using the SR50 mode.

RNA-seq data (FASTQ files) were aligned to the hg38 reference genome using STAR (v2.5.4b). Ensembl annotation data GRCh38.84 were used in the alignment process (--sjdbGTFfile option). featureCounts from the Subread package (v2.0.0) was used to count the number of features (uniquely mapped reads) over the exons (-t exon, as default) summarized by gene ID (-g gene_id). The percentage of uniquely mapped reads assigned to features was around 80%. The output raw count matrices (genes-by-samples) were used as input data for the analysis, performed with the R package DESeq2 (v1.24.0). We had eight samples and four conditions (NM-scr, HT-scr, HT-LNA1, and HT-LNA2), each condition with two biological replicates. Biological replicates were not merged together in the analysis, whilst they were considered as a single condition for differential expression analysis. As a first step, prefiltering was performed to remove genes with very low counts. We performed a minimal prefiltering (default in DESeq2) by removing genes with less than ten reads total across all the eight samples. From a total of 60,675 genes, we obtained 20,144 genes after filtering that were used for the following analysis. Then, we performed the standard differential expression analysis of DESeq2. Before performing the analysis, counts were normalized using the median-of-ratios method, to take into account for sequencing depth and RNA composition (possible presence of a few highly differentially expressed genes between samples, differences in the number of genes expressed between samples, presence of contamination, etc.). Gene length does not need to be considered given that the analysis is performed comparing the normalized counts for the same gene between conditions. After normalization, differential expression analysis was performed using the Wald test with HT-LNA1 as reference condition. To call differentially expressed genes, we used a threshold of 0.05 on the BH (Benjamini–Hochberg) adjusted $p$ values. Heatmap in Fig. 4d represents the $z$-scaled gene expression levels and was generated as follows: (1) normalized counts per each condition were calculated by averaging the normalized counts across the two biological replicates; (2) normalized counts were log-transformed as $\log 2(x + 1)$; and (3) $z$-scores for the four conditions were calculated using the log-transformed values.

**DNA FISH experiment and imaging analysis.** DNA FISH probes were obtained from Empire Genomics, LLC. and tested for hybridization specificity. Information of DNA FISH probe design is listed in Supplementary Fig. 7. Probes used for DNA FISH were designed against genomic regions overlapping LINC00607-SERPINE1 chimeric read aligned regions. Region of chromosome 2 that exhibited strong hybridization signal in FISH, but do not show significant interacting signal with *Serpine1SE*, was used as the control.

DNA FISH was performed by following probe manufacturer protocol (Empire genomics) with minor modifications. Briefly, for imaging experiment, HUVEC cells were seeded onto poly-L-Lysine pre-coated coverslips (Fisher Scientific, #NC0326897), and treatment was performed as described. All incubation steps were performed at RT unless otherwise indicated. On Days 3 and 7 of treatment, cells were washed twice in PBS and fixed in fresh 4% paraformaldehyde (pH 7.2) for 30 min.

Cells were permeabilized in 0.1% saponin (Sigma-Aldrich, #84510), 0.1% Triton X-100 in PBS for 10 min at RT. Subsequently, cells were incubated in 20% glycerol in PBS for 20 min, followed by three repetitions of freeze-thaw in liquid nitrogen. The slides were then denatured in 0.1 M HCl for 30 min, and blocked in 3% BSA and 100 μg/mL RNase A in PBS for 1 h at 37 °C. This was followed by a second permeabilization step in 0.5% saponin/0.5 % Triton X-100 in PBS for 30 min. The slides were then denatured sequentially in 70% formamide (Invitrogen, #AM9342)/ 2× SSC (Sigma-Aldrich, #S6639) for 2.5 min, then 50% formamide/2× SSC for 1 min in 73 °C water bath, before immediately incubated with probe mixtures denatured at 75 °C for 5 min. Probes were prepared according to manufacturer protocol. After 18 h of hybridization in a dark humid chamber, the slides were washed with agitation in the following solutions sequentially: 50% formamide/2× SSC, 2× SSC (37 °C for both), and 4× SSC/0.1% Tween 20. DAPI staining and slides mounting (Thermo Fisher Scientific, #00-4958-02) were performed after PBS rinse.

Slides were imaged using Perkin Elmer UltraView Vox Spinning Disk Confocal 63× oil immersion objective lens. Distance between red and green fluorescent FISH signal spots in the nucleus was quantified using Fiji (http://fiji.sc/wiki/index.php/Fiji) and Matlab software. $P$ values were generated using nonparametric Wilcoxon test with Bonferroni correction for multiple comparisons. Statistical tests were performed in R.

**RNA FISH experiment and imaging analysis.** LNA GapmeR probes from QIAGEN were used. Sequence of RNA FISH probe targeting LINC00607 is listed in Supplementary Table 6. HUVEC cells were seeded onto poly-L-Lysine pre-coated coverslips and treated with mannitol or HG + TNFα for 3 days. Cells were washed twice in PBS and fixed in fresh 3.7% formaldehyde (pH 7.2) for 10 min. Cells were permeabilized in 70% ethanol at least overnight at 4 °C. Slides were incubated with 1 mL of wash buffer A at RT for 2–5 min. Then the slides were incubated with LNA-probes 1:100 dilute in the hybridization buffer (Biosearch Technologies). After 16 h of hybridization in a dark humid chamber at 37 °C, the slides were washed with wash buffer A (Biosearch Technologies) in the dark for 30 min. Following washes, the cells on the slides were stained with DRAQ5 (Abcam, AB108410, 1:1000 dilution) in dark for 30 min. The slides were incubated with wash buffer B (Biosearch Technologies) at RT for 2–5 min and were imaged on a Leica SP5 Confocal microscope using 63×/1.40 oil immersion objective lens. Fluorescent FISH signal was captured using Leica LAS AF software in 1024 × 1024 format. All images were taken using the same power, gain, and collection bands for respective fluorophores to allow equal comparison of fluorescence levels of samples.

**Subcellular fractionation and RNA isolation.** Subcellular fractionation was performed using HUVECs from three confluent 100 mm culture dishes as independent triplicates. The cells were collected, washed in 5 mL cold PBS, and centrifuged at 300 × g for 5 min at 4 °C. The cell pellets were lysed on ice for 10 min in 500 μL cold cytoplasmic lysis buffer (10 mM HEPES, 1.5 mM MgCl$_2$, 10 mM KCl, 0.5 mM DTT, 0.05% NP-40, and protease and RNase inhibitors; pH 7.9) and then centrifuged at 1811 × g in a swing bucket centrifuge at 4 °C. After separating the supernatant containing the cytosolic fraction, 3× volume of TRIzol LS was added immediately for RNA extraction. The pellet containing the nuclear fraction was gently resuspended in 400 μL cold nuclear buffer (5 mM HEPES, 1.5 mM MgCl$_2$, 300 mM NaCl, 0.2 mM EDTA, 0.5 mM DTT, 26% glycerol (v/v), and protease and RNase inhibitors; pH 7.9), homogenized using a douncer, and lysed on ice for 30 min. Following the lysis, the nuclear fraction was centrifuged at 20,000 × g for 30 min at 4 °C. The supernatant containing the nucleoplasmic fraction was mixed immediately with 3× volume of TRizol LS for RNA extraction. The remaining pellet containing the chromatin fraction was resuspended in 50 μL of cold PBS and used immediately for RNA extraction using TRIzol. The RNA extracted from the three different fractions was dissolved in 20 μL of RNase-free water and equal volumes of RNA were used for reverse transcription and qPCR.

**Monocyte adhesion assay.** Deidentified human peripheral blood mononuclear cells (PBMCs) were obtained through leukocyte reduction system chambers/cones routinely collected at City of Hope after plateletpheresis. The research consents were obtained from the donors' next of kin and ethical approval for this study was granted by the Institutional Review Board of City of Hope (IRB #09025). PBMCs

were isolated by density gradient centrifugation over Ficoll-Paque (GF Healthcare) using SepMate™ tubes (STEMCELL Technologies Inc.) according to the protocol provided by the manufacturer. Subsequently, monocytes were isolated from PBMCs by immunomagnetic positive selection of CD14+ cells using CD14 microbeads (Miltenyi Biotec). Monocytes adhesion assay was performed using the isolated monocytes from four different healthy donors, which were labeled with CellTracker™ Green CMFDA Dye (Thermo Fisher Scientific) and incubated with monolayer ECs ($4 \times 10^3$ cells per cm$^2$) for 15–30 min in a cell culture incubator. The nonattached monocytes were then washed off with complete EC growth medium. The attached monocyte numbers were evaluated on Cytation™ 1 Cell Imaging Multi-Mode Reader (BioTek) using green fluorescent channel. Average numbers per sample were calculated from five randomly selected fields.

**SA-β-gal staining**. Cytochemical staining for SA-β-galactosidase was performed using the Senescence β-Galactosidase Staining Kit (Cell Signaling Technology) following the manufacturer's manual. Briefly, the ECs post transfection and H + T treatment were washed once with freshly prepared 1× PBS, fixed in 1× fixative solution for 10–15 min at RT, and then rinsed twice with 1× PBS. The cells were stained for 48 h in a dry incubator before viewing under a Leica DMi1 microscope (Leica microsystems) at 50× magnifications. The percentage of SA-β-galactosidase-positive cells was determined by counting the number of blue cells under bright field illumination.

**Statistical analysis**. For all experiments in Figs. 1 and 4, at least three independent experiments were performed unless otherwise specified. Statistical analysis was performed using Student's $t$ test (two-sided) between two groups or ANOVA followed by Bonferroni post-test for multiple-group comparisons. $P < 0.05$ was considered as statistically significant. For other experiments, the quantification and statistical analysis have been specified and detailed in the results, figure legends, and methods.

**Reporting summary**. Further information on research design is available in the Nature Research Reporting Summary linked to this article.

## Data availability
All high-throughput data supporting the current study are accessible on GEO under accession number GSE13535. Ensembl annotation data GRCh38.84 (Homo_sapiens.GRCh38.84.gtf.gz) are publicly available at ftp://ftp.ensembl.org/pub/release-84/gtf/homo_sapiens/Homo_sapiens.GRCh38.84.gtf.gz. HUVEC SE data (HUVEC.bed) were downloaded from dbSUPER (https://asntech.org/dbsuper/data/bed/hg19/HUVEC.bed). HUVEC enhancer data (HUVEC.fasta) were downloaded from EnhancerAtlas (http://enhanceratlas.org/data/enhseq/HUVEC.fasta). Other data are available from the corresponding authors upon reasonable request. Source data are provided with this paper.

## Code availability
The codes used for the analysis have been deposited and made publicly available on GitHub at https://github.com/Zhong-Lab-UCSD/NCOMMS-19-24818.

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

## Acknowledgements

This work was supported by NIH grants R00HL122368, R01HL121365, and R01HL108735 (to Z.B.C.); DP1DK126138, DP1HD087990, and U01CA200147 (to S.Z.); R01DK065073, R01DK081705, and R01HL106089 (to R.N.); an Ella Fitzgerald Foundation grant and a Wanek Family Project (to Z.B.C.); and a Human Cell Atlas seed network grant (to Z.B.C. and S.Z.). Research reported in this publication included work performed in the Integrative Genomics Core at City of Hope supported by the National Cancer Institute of the National Institutes of Health under award number P30CA033572. The authors would like to thank Drs. Ismail Al-Abdullah and Meirigeng Qi of the islet transplantation team at City of Hope for isolation of human tissues, Drs. Saul Priceman and Yukiko Yamaguchi at City of Hope for the generous gifts of human monocytes, and Ms. Aleysha Chen at University of California Berkeley for her excellent scientific editing.

## Author contributions

Conceptualization, Z.B.C. and S.Z.; methodology, Z.B.C., S.Z., R.C., L.X., Y.L., W.W., X.F., A.B.B., X.T., C.C., and T.N; investigation and validation, L.X., Y.L., R.C., K.S., X.F., C.C., and T.N.; writing-original draft, Z.B.C., R.C., L.X., and S.Z.; writing-review and editing, Z.B.C., S.Z., R.C., R.N., and Y.L.; resources, Z.B.C, S.Z., and R.N.; supervision, Z.B.C. and S.Z.; and funding acquisition, Z.B.C and S.Z..

## Competing interests

S.Z. is a founder and board member of Genemo, Inc. The other authors declare no competing interests.
