## [Peer Review File · Nature Communications]

Reviewers' comments:

Reviewer #1 (Remarks to the Author):

Calandrelli et al., present elegant technical in vitro work with potential novelty on the contribution of RNA-chromatin interactions at superenhancers (SEs) in promoting endothelial dysfunction. Endothelial dysfunction underlies the development of multiple diseases states and the authors aim to understand the genomic and molecular mechanisms of this process. They postulated that long range RNA-DNA interactions at super enhancers (SE) could be engaged in endothelial dysfunction and in endothelial to mesenchymal transition (EndoMT). The authors used a cell model of endothelial dysfunction using HUVECs treated with high glucose (H,25mM) and TNF α (T, 5ng/ml) and showed reduction of eNOS and induction of α -SMA, markers of endothelial dysfunction. Employing single-cell RNA seq and iMARGI, their prior published bioinformatic approach, they provide evidence for inter-chromosomal RNA-chromatin interactions as opposed to intra-chromosomal interactions, especially among SEs, in underlying the endothelial dysfunction by H+T treatment. They interrogate the mechanism of novel lincRNA, linc00607 SE and its interaction with SERPINE1/ PAI-1 SE and provide some evidence of a novel interaction of linc00607 and SERPINE1/ PAI-1 and its contribution to endothelial dysfunction. The work has novel features and is of potential interest to the broader scientific community if it can convincingly provide evidence of RNA-DNA inter-chromosomal interactions at SEs that could explain how genomic loci at different chromosomal locations are activated in a coordinated fashion in endothelial dysfunction, and perhaps many important cell biological functions and pathophysiological settings. However, there are substantial concerns that should be addressed.

Major concerns:

1. Novelty is claimed in showing “a novel molecular mechanism at the RNA-chromatin contact level underlying their reported central role in EC dysfunction and vascular diseases” – based on “dynamic and sustained activation of these SE hubs and the associated SE networks suggests”. Yet (as authors do note), this is not altogether novel as (1) inter- and trans- chromosomal RNA-DNA interactions have been established as causal drivers of multiple cell specific biologies, (2) previous study demonstrates robust activation of inflammatory SEs in ECs transiently treated with TNF α , and (3) emerging data/refs (eg 46/47) supporting key roles for RNA-chromatin interactions. So a stronger case should be made for the specific novelties – based on more detailed evidence and more examples too perhaps.
2. What is the evidence for directionality or sequence of causal events proposed in discussion? (“implicating that RNA-chromosomal interactions may form due to the activation of SEs, which may in turn promote inter-chromosomal DNA-DNA interaction in a cell-type- and stimulus-specific manner.”) Further, the fig 3 data is striking and perhaps surprising - “Based on the Hi-C data, there was no significant change among either the intra- or inter-chromosomal interactions (Fig. 3)” – is this novel, expected, demonstrated previously, is it generalizable in ECs and other cells? Independent validations should be provide.
3. The induced RNA-DNA SE interaction example (LINC00607-SERPINE1 SE) is quite interesting – but more work needs to be done on the bioinformatic and molecular level with LINC00607. This includes basic information including cell-specificity, EC regulatory features at lncRNA, exonic structure and isoform data, screening for translation (proteins and micro-peptides by lncRNA) etc. Given apparent

cytoplasmic expression, other functions of LINC00607 should be considered beyond that studied (RNA-DNA interactions between LINC00607 and SERPINE1 SE). The LNA1 vs. LNA2 data is not strong proof and not convincing for an exclusive function “supporting the role of LINC00607 and its engaged SE network in promoting endothelial dysfunction”. Functions in cis are mentioned and these are likely distinct from those related to induced inter-chromosomal LINC00607 and SERPINE1 SE. Is there evidence that LINC00607 interacts with specific DNA elements in nucleus or proteins in cell (nucleus or cytoplasm)? Knock-down (KD) by one strategy (LNAs), especially with differential effects of LNA1 vs. 2, lacks rigor and orthogonal complementary. Alternative approaches including overexpression (OE) of LINC00607 should be presented – and studies should include isoform-specific KD and OE if multiple isoforms exist in endothelial cells.

4. In this context and in context of multiple induced SE hubs (1 going to 25), more induced RNA-chromatin interactions should also be presented and validated to support the generalizability and claim of novelty for novel molecular mechanism at the RNA-chromatin contact level underlying role in EC dysfunction and vascular diseases.

5. The endothelial cell dysfunction model warrants more support – particularly more physiological/pathophysiological and in vivo or ex vivo data. The glucose concentration is very high to be considered pathophysiological relevant in most disease setting. Confirmation and validation at lower concentrations or profiling of alternative stress models should be presented. Replication in arterial ECs is warranted. In the absence of in vivo experimental models, validation in ex vivo human EC samples would be of great value. Is this truly an EndoMT model – does the low level of SMA in HUVEC at day 7 (1.4% of cells) support this?

6. The authors might elaborate more on the biological and translational significance and generalizability and potential impact of findings, in their concluding paragraph?

Minor comments:

1. Why do the authors expect specific directionality in the assumption here; “more DE genes upregulated than downregulated by Day 7 of H+T treatment (Fig. 2c), we argued that the increased inter-chromosomal RNA-chromatin interactions, rather than the decreased intra-chromosomal interactions, would be more directly related to the transcriptional induction of genes promoting EC dysfunction.”

2. What is the experimental evidence to support the endothelial SEs dataset (ref 30, dbSUPER) selected for endothelial (and what cell types)?

3. How many lncRNA detectable in these HUVEC experiments with 10G (speaks to low sensitivity/missing data) – e.g., compared to bulk-seq?

4. Fig 1. CD31 not expressed on most HUVECs at baseline – this is

5. Figure 1b- Negative control bars do not have an error bar? Why was control set as 1 for both N+M and NT?

6. Fig 6b-c. Control data for lncRNA looks very noisy – please address.

7. Page 13 “The inhibitory effect of LINC00607 LNA1 in monocyte adhesion to ECs was consistent using peripheral blood-derived monocytes from 4 different donors (Fig. 7d).” Figure 7d only shows 3 donors, change 4 to 3 in text.

8. Fix apparent error in sentence. “However, at the molecular level, whereas the H+T suppressed eNOS and ICAM1 mRNA levels could be almost completely reversed, those of LINC00607, SERPINE1, FN1, and CTGF could not be reversed. In fact, the induction of LINC00607, SERPINE1, and FN1 continued despite the removal of detrimental stimuli (Supplemental Fig. 10b).” H+T increased ICAM mRNA levels not suppressed. Also, are there any statistics for this figure?

9. Labeling of some figures should be improved: Figure 6a is missing Day 0, 3 and 7 above SE hub maps – can this figure be made bigger: Authors should refer to the names used in the figures for ease of reading e.g., ACTA2 in figure 2E - add ACTA2 in brackets after α -SMA: Also eNOS/ NOS3 does not appear in the heatmap of figure 2D; Clarify which figure is referred to here “Page 11, line 3 ‘Given this definition, there was only 1 SE hub (which overlaps with MALAT1) identified in control (Day 0) ECs, with 131 active SE nodes and 130 interacting SE pairs.’

10. The authors should define the statistics used and definitions for *, **, *** in each of their figure legends? They should also include “n” numbers and experimental replicates for each experiment in every figure legend.

11. The authors should provide a figure to show expression of LINC00607 in the cytoplasm and nucleus – both by fractionation and smFISH?

Reviewer #2 (Remarks to the Author):

Review of “Dynamic changes in RNA-chromatin interactome promote endothelial dysfunction”

In this highly innovative paper, Calandrelli and colleagues used a newly developed technology (called iMARGI) to study the RNA-chromatin interaction landscape changes during endothelial to mesenchymal transition (EndoMT) and impact on gene regulation. They found dramatic changes mainly characterized by marked increase of inter-chromosomal interactions and formation of new chromatin hubs. Interestingly, the Hi-C profiles show little changes. A key observation is that these hubs are enriched with super-enhancers, indicating the dynamic landscape of RNA-chromatin interactions may play a major role in gene regulation. They further showed that lncRNAs (with SERPINE1 as an example) could be a major player driving the RNA-chromatin dynamics. Altogether, the paper has discovered a novel gene regulatory mechanism through lncRNA mediated inter-chromosomal interactions which may have important consequences in diseases. The data are convincing and the paper is very well-written.

I have a few minor questions/suggestions for the authors.

1. In Hi-C experiments, it is known to be difficult to identify inter-chromosomal interactions. I wonder if this is also the case here, and if so is there any way to determine the robustness of chromatin hub identification.

2. The model that lncRNAs mediated RNA-chromatin interactions could regulate gene expression by mediating SE formation is very interesting, but I think the argument could be made stronger if additional tests could be done to measure the degree of RNA-chromatin interactions changes due to lncRNA perturbation. Of particular interest is if the SE chromatin state is lost. The reported evidence here seems

to be somewhat indirect.

3. The single-cell RNAseq data does not seem to be fully utilized in the paper. The authors might consider to correlate the variation of lncRNA and target gene expression levels which would offer additional support to their model. i.e, the expression of a few key lncRNAs is required for activating SE associated genes. Also, the tSNE plot (Figure 2b) seems to suggest that the cell states at the three time points are completely different. Would it be possible that some of these differences may be due to technical variation such as batch effects?

4. In a broader context, the concept that chromatin hub plays a role in SE is not new, for example, see Huang et al. Nat Commun. 2018 Mar 5;9(1):943. It would be nice to acknowledge such previous work.

5. A list of SE hubs should be provided as a supplementary table.

Reviewer #3 (Remarks to the Author):

Calandrelli et al utilize a technique (iMARGI) that can map global RNA-genome interactions to understand how endothelial cells respond to stimuli that induce inflammation and endothelial-to-mesenchymal transition (endoMT). Surprisingly, global DNA-DNA interactions, as mapped by HiC did not change following exposure of endothelial cells to glucose and TNF-alpha for 7 days, despite major changes in cellular phenotype and gene expression programs (assessed by single-cell RNA-seq). Instead, they found that RNA-DNA interactions were highly altered, especially inter-chromosomal interactions. Many of these interactions appeared to preferentially occur in super-enhancer loci. While intra-chromosomal interactions decreased, inter-chromosomal interactions increased with glucose + TNF-alpha treatment. They examined one of these inter-chromosomal loci in detail (SERPINE1-LINC00607) and found that the lncRNA regulates SERPINE1 expression and several other inflammatory/endoMT genes as well as the inflammatory and endoMT phenotype of endothelial cells. The findings presented in this manuscript are highly novel and interesting. The data will be a rich resource of data for the community. However, several questions remain.

Major Comments:

1) The discovery of new inter-chromosomal hubs of RNA-DNA interactions at super-enhancer loci in response to glucose/TNF-alpha stimulation, including between the SERPINE1-LINC00607 loci, is intriguing. However, additional functional data connecting LINC00607 to the regulation of the hub(s) that it interacts with is required. Does knock-down of LINC00607 affect the proximity of the SERPINE1 locus and the LINC00607 locus? Are the genes dysregulated by LINC00607 knock-down preferentially in the areas of interaction between LINC00607 and the genome? This could be assessed by RNA-seq. The current manuscript only has a few select genes assessed by qRT-PCR. Does LINC00607 regulate the establishment of enhancer marks in the genomic regions that it interacts with?

- 2) It would be helpful to include additional lncRNA-DNA interaction maps for comparison to LINC00607. This would emphasize that the interaction with the SERPINE1 locus is unique for this lncRNA. Is the interaction between LINC00607 RNA and the SERPINE1 locus in Fig. 6C significantly above background?
- 3) The imaging analysis of the endoMT phenotype could be improved. Confocal imaging of VE-Cadherin and co-staining with Phalloidin (to show cell shape change) would be helpful. The brightfield images provided are difficult to interpret and the CD31 staining was very patchy.
- 4) Are genomic DNA-DNA interactions between the SERPINE1 and LINC00607 loci observable from HiC data and do they change during the time-course of treatment?
- 5) Is there an independent method that can be used to validate some of the RNA-DNA interactions? The finding of such widespread inter-chromosomal interactions is very intriguing, but additional independent evidence would be helpful. This is especially the case if only one replicate was used for analysis. The number of replicates should be clearly stated in the paper. Does LINC00607 RNA FISH reveal co-localization with the SERPINE1 locus?
- 6) Is LINC00607 enriched in the nucleus? This data should be available from the raw data used in Fig. 7B.
- 7) The data should be made available in a public data repository.

Point-to-point response

Reviewer #1:

Calandrelli et al., present elegant technical in vitro work with potential novelty on the contribution of RNA-chromatin interactions at superenhancers (SEs) in promoting endothelial dysfunction. Endothelial dysfunction underlies the development of multiple diseases states and the authors aim to understand the genomic and molecular mechanisms of this process. They postulated that long range RNA-DNA interactions at super enhancers (SE) could be engaged in endothelial dysfunction and in endothelial to mesenchymal transition (EndoMT). The authors used a cell model of endothelial dysfunction using HUVECs treated with high glucose (H,25mM) and TNF α (T, 5ng/ml) and showed reduction of eNOS and induction of α -SMA, markers of endothelial dysfunction. Employing single-cell RNA seq and iMARGI, their prior published bioinformatic approach, they provide evidence for inter-chromosomal RNA-chromatin interactions as opposed to intra-chromosomal interactions, especially among SEs, in underlying the endothelial dysfunction by H+T treatment. They interrogate the mechanism of novel lincRNA, linc00607 SE and its interaction with SERPINE1/ PAI-1 SE and provide some evidence of a novel interaction of linc00607 and SERPINE1/ PAI-1 and its contribution to endothelial dysfunction.

The work has novel features and is of potential interest to the broader scientific community if it can convincingly provide evidence of RNA-DNA inter-chromosomal interactions at SEs that could explain how genomic loci at different chromosomal locations are activated in a coordinated fashion in endothelial dysfunction, and perhaps many important cell biological functions and pathophysiological settings. However, there are substantial concerns that should be addressed.

We appreciate the reviewer's comments on the technical elegance and novelty of our study. As follows, we tried our best to address the reviewers' concerns.

Major concerns:

1. Novelty is claimed in showing "a novel molecular mechanism at the RNA-chromatin contact level underlying their reported central role in EC dysfunction and vascular diseases" – based on "dynamic and sustained activation of these SE hubs and the associated SE networks suggests". Yet (as authors do note), this is not altogether novel as (1) inter- and trans- chromosomal RNA-DNA interactions have been established as causal drivers of multiple cell specific biologies, (2) previous study demonstrates robust activation of inflammatory SEs in ECs transiently treated with TNF α , and (3) emerging data/refs (eg 46/47) supporting key roles for RNA-chromatin interactions. So a stronger case should be made for the specific novelties – based on more detailed evidence and more examples too perhaps.

We appreciate the reviewer's comments. To strengthen our novelties, we have addressed the three points raised:

- 1) To the best of our knowledge, there have only been a handful of studies that have investigated inter- and trans- chromosomal RNA-DNA interactions, including one from

us (Miao *et al. Nat Commun* 9: 292, 2018; Hacısuleyman, et al. *Nat Struct Mol Biol* 21: 198-206, 2014.]. These studies have focused on individual chromatin-associated RNAs (caRNAs), but have not interrogated the global RNA-chromatin contacts in a genome-wide fashion.

- 2) Our study is not to re-emphasize the importance of SEs *per se* but to unravel the importance of interactions between SEs and se-caRNAs in transcriptional regulation. Compared to studies investigating SEs and DNA-DNA interactions, the RNA-chromatin contacts, especially inter-chromosomal RNA-DNA interactions have been very little studied at a genome-wide scale due to the limitation of pertinent technologies. The previous study from Brown *et al. Mol Cell* 2014 identified the activation of inflammatory SEs in ECs transiently treated with TNF α (1-4 hr) and specifically focused on the binding by transcription factor and coactivator. This study did not address the role of RNAs nor their interaction with SEs in the transcriptional induction of inflammation.
- 3) Although the pioneering papers (refs 46/47) highlighted the potential importance of caRNAs based on the observation of pervasive RNA attachment to chromatin in unperturbed cells, there has been no study to assess the genome-wide changes in RNA-chromatin interactions in cells undergoing dynamic biological and functional changes (exemplified by EC dysfunction in our study).

Taken together, the novelty of our study comprises: 1) revealing the dynamics of RNA-chromatin interactomes in a disease-driven process and its association with genomic interactions and transcriptome changes; 2) identifying that the activation of SE-involved RNA-DNA interactions (esp. inter-chromosomal interactions) contributes to transcriptional induction promoting EC dysfunction, supporting the inter-chromosomal RNA-activation hypothesis. We have revised our manuscript to clarify and strengthen the novelties of our study in the Introduction and Discussion.

2. *What is the evidence for directionality or sequence of causal events proposed in discussion? (“implicating that RNA-chromosomal interactions may form due to the activation of SEs, which may in turn promote inter-chromosomal DNA-DNA interaction in a cell-type- and stimulus-specific manner.”) Further, the fig 3 data is striking and perhaps surprising - “Based on the Hi-C data, there was no significant change among either the intra- or inter-chromosomal interactions (Fig. 3)” – is this novel, expected, demonstrated previously, is it generalizable in ECs and other cells? Independent validations should be provide.*

Indeed, we do not have direct evidence for the suggested directionality/sequence of these causal events as previously noted in the discussion. We have removed this statement from the revised manuscript.

The lack of significant change in genomic interactions in the presence of dramatic transcriptional change has not been demonstrated in ECs previously. This is striking but not entirely surprising. Conceptually, even though ECs undergo dramatic changes in transcriptional state and cellular function during the H+T treatment, they remain largely ECs (as shown in our scRNA-seq data, Fig. 1c). In line with a recent study (Ray *et al., PNAS* 116: 19431-19439, 2019) showing that topologically associating domains (TADs) and compartment structures remain unchanged upon acute heat shock in human and *Drosophila*, while only modest changes of distal regulatory

interactions are observed in human cells, our Hi-C data revealed lack of significant changes in TADs and compartments in H+T-treated ECs (Supplementary Fig. 3). However, the observation that Hi-C data did not reveal any significant changes in DNA-DNA interactions whereas RNA-DNA interactions show profound global changes in a given cellular context has not been demonstrated previously. While it is interesting for future studies whether such phenomenon is generalizable in ECs under other stimuli and in other cells, we consider this out of the current scope of the study.

3. *The induced RNA-DNA SE interaction example (LINC00607-SERPINE1 SE) is quite interesting – but more work needs to be done on the bioinformatic and molecular level with LINC00607. This includes basic information including cell-specificity, EC regulatory features at lncRNA, exonic structure and isoform data, screening for translation (proteins and micro-peptides by lncRNA) etc. Given apparent cytoplasmic expression, other functions of LINC00607 should be considered beyond that studied (RNA-DNA interactions between LINC00607 and SERPINE1 SE). The LNA1 vs. LNA2 data is not strong proof and not convincing for an exclusive function “supporting the role of LINC00607 and its engaged SE network in promoting endothelial dysfunction”. Functions in cis are mentioned and these are likely distinct from those related to induced inter-chromosomal LINC00607 and SERPINE1 SE. Is there evidence that LINC00607 interacts with specific DNA elements in nucleus or proteins in cell (nucleus or cytoplasm)? Knock-down (KD) by one strategy (LNAs), especially with differential effects of LNA1 vs. 2, lacks rigor and orthogonal complementary. Alternative approaches including overexpression (OE) of LINC00607 should be presented – and studies should include isoform-specific KD and OE if multiple isoforms exist in endothelial cells.*

We have now provided basic information regarding LINC00607 as follows:

Cell-specificity and EC regulatory features:

Based on RNA-seq and RNA-polymerase 2 (Pol2)-ChIP-seq data available at ENCODE, LINC00607 is highly expressed in human ECs. It is also expressed in dermal fibroblasts but at lower level, and in embryonic stem cells at even lower level. In several other cell types, including epithelial cells, skeletal myotubes, embryonic kidney and lung fibroblasts, and monocytes, LINC00607 is expressed at marginal or undetected levels. Accordingly, only in HUVECs *LINC00607* TSS show strong enrichment for POL2 binding and the entire region, particularly the exons show strong H3K27ac signals (Figure

Regulatory features of LINC00607 based on RNA-seq and ChIP-seq data.

to the right).

Exonic structure and isoform data:

According to LNCipedia (version 5.2), LINC00607 has 7 isoforms/transcript variants, and 6 of them are annotated in Ensembl with exon structures (see the figure and table below). Among these transcripts, only transcript 3 (LINC00607:3) is validated in the NCBI Refseq (NR_037195.1). Based on Encode data, LINC00607:3 is the most abundant LINC00607 transcript and localized to the nucleus in HUVECs. It was used as the main transcript for qPCR detection and LNA design in our study.

Illustration of exon arrangement of LINC00607 isoforms. Rectangle shapes represents exons.

Expression of LINC00607 transcript variants in HUVECs

Transcript ID	Size (nt)	# of Exons	Relative Expression in HUVECs (ENCODE)								
			Dataset 1: Whole cell (RPKM)		Dataset 1: Cytosol (RPKM)		Dataset1: Nucleus (RPKM)		Dataset 2 (FPKM)		
			rep1	rep2	rep1	rep2	rep1	rep2	rep1	rep2	
LINC00607 :1	18176	12	NA	NA	NA	NA	NA	NA	NA	NA	NA
LINC00607 :2	558	5	0.06	0.05	0.04	0.08	0	0.07	7.5	7	
LINC00607:3	3690	10	5	5.04	4.41	3.85	15.51	14.24	16.78	25.86	
LINC00607 :4	581	5	0.21	0.81	0.16	0.15	0.15	0.14	11.46	4.74	
LINC00607 :5	570	5	0.19	0.12	0.11	0.11	0.1	0.09	13.02	22.64	
LINC00607 :6	2104	9	1.8	2.07	2.45	2.03	4.2	5.52	17.25	19.01	
LINC00607 :7	526	2	2.63	2.14	1.96	3.81	3.57	3.38	0.47	0.69	

Screening for translation potential:

We tested the coding potential of LINC00607 using Coding-Potential Assessment Tool (CPAT) (Wang *et al Nucleic Acids Res.* 41: e74. 2013) and found that none of the LINC00607 transcripts

have any coding potential (table below). We used LEENE, an EC lncRNA that we previously studied (Miao *et al. Nat Commun* 9: 292, 2018) and EGFR (a protein-coding genes) as controls.

Transcript name/ Ref Seq ID	Transcript size (nt)	ORF size (nt)	Ficket Score	Hexamer Score	Coding Probability	Coding Potential
LEENE NR_026797.1	2030	258	0.696	-0.21727	0.00921	No
EGFR NM_001346941.2	9104	2832	1.0585	0.33157	1.00000	Yes
LINC00607:1	18176	354	0.5783	-0.17354	0.00900	No
LINC00607:2	558	198	0.5934	0.04319	0.02103	No
LINC00607:3 NR_037195.1	3690	249	0.8413	-0.03280	0.03909	No
LINC00607:4	581	255	0.546	-0.01374	0.02297	No
LINC00607:5	570	315	0.7654	0.05896	0.12656	No
LINC00607:6	2104	228	0.7365	-0.06916	0.01980	No
LINC00607:7	526	105	1.1688	-0.15792	0.01257	No

Consistently, ribosomal profiling in 43 publicly available datasets on Encode revealed insignificant signal in *LINC00607* region (figure below). Together, these data suggest a lack of translational potential of *LINC00607*.

Ribo-seq signals at *LINC00607* genomic region based on 43 datasets visualized on GWIPS-viz Genome Browser

We have also summarized the above-mentioned basic information in the newly added Supplementary Fig. 6.

As emphasized in our revised manuscript, the focus of this study is to identify the regulatory role of caRNAs and inter-chromosomal RNA-chromatin interactions in transcription. We used *LINC00607* as an example of SE-derived caRNA but we do not intend to prove its exclusive

function in any subcellular compartment, or an exclusive role in its engaged SE. We agree with the reviewer that the functions *in cis* are likely distinct from those related to induced inter-chromosomal LINC00607 and SERPINE1 SE, and thus have removed the mentioning of *cis* regulation by LINC00607.

Regarding whether there is “evidence that LINC00607 interacts with specific DNA elements in nucleus or proteins in cell (nucleus or cytoplasm)”, there is no evidence that LINC00607 interacts with specific DNA elements in nucleus or proteins in the cell.

To complement the experiments using LNA, we have taken the reviewer’s excellent suggestion and performed LINC00607 OE experiments. Specifically, we overexpressed LINC00607:3 in HUVECs. Based on 6 independent repeats of the experiments, we did not observe any consistent effect of LINC00607 on SERPINE1 expression (figure below). These data suggest that LINC00607 alone is insufficient to alter the transcription of its interacting targets. We have added this data in the Supplementary Fig. 9.

Effect of LINC00607 OE in SERPINE1. (Left) LINC00607:3 was cloned into a pcDNA3.1(+) vector. HUVECs were transfected with either empty pcDNA3.1(+) plasmid or plasmid with LINC00607 cDNA (0.6 μ g per well in 6-well plates). Cells were harvested at 48 hr post transfection. RNA levels of respective genes were quantified by qPCR (left panel). Data are represented as mean \pm SEM from 6 independent experiments. * p < 0.05 compared with respective control (empty vector of the same dosage). (Right) Subcellular fractionation followed by qPCR was performed to confirm the overexpressed LINC00607 in the nucleus and chromatin (right panel). MALAT1 was detected as a positive control.

4. In this context and in context of multiple induced SE hubs (1 going to 25), more induced RNA-chromatin interactions should also be presented and validated to support the generalizability and claim of novelty for novel molecular mechanism at the RNA-chromatin contact level underlying role in EC dysfunction and vascular diseases. LINC01013/02154 increased interaction with any SE (hubs)?

Taking the reviewer’s outstanding suggestion, we have done the following:

1) We have presented more induced RNA-chromatin interactions for both the emergent hub SE networks and *Linc607SE*-interacting networks (comprising 12 SEs). Please refer to the new Figs. 3a and 5a.

2) We have also performed RNA-seq using ECs with LINC00607 knockdown and examined the inhibitory effect on the expression of genes embedded in other SEs showing induced RNA-chromatin interactions with *Linc607SE*. RNA-seq revealed that in addition to SERPINE1, the expression of 13 other genes contained in 12 SEs was also suppressed by LINC00607 knockdown (KD) (figure to the right; also see Supplementary Table 5).

RNA-seq data from ECs with LINC00607 KD. HUVECs were transfected with scramble (scr) or LNA inhibiting LINC00607 before subjected to mannitol (NM) or H+T (HT) as indicated. Heatmap plotted with Z score.

3) We have taken an alternative approach for testing the emergent regulatory link between LINC00607 and genes contained in all the *Linc607SE*-interacting SEs using scRNA-seq data from H+T-treated ECs as well as diabetic donor-derived ECs (Fig. 5, b and c; also included below). The correlation analyses suggest that the SEs exhibiting H+T-induced interactions are more likely to co-express their harbored genes in the same single cells in dysfunctional ECs than in healthy ECs. Furthermore, the data from cultured ECs and donors' ECs show a high degree of consistence.

(Figure 5) Increased single-cell co-expression of LINC00607 and SERPINE1 in dysfunctional vascular endothelium. (A) H+T induced interactions (edges) between Linc607SE (yellow node) and hub SEs (red nodes). (B-C) Odds ratio (y axis) between single-cell co-expression levels and the health status of ECs, including H+T vs. control ECs (B) and diabetic (T2D) vs. healthy vascular endothelium (C). Each dot corresponds to a gene in any Linc607SE-interacting SE (panel A). A large odds ratio corresponds to a positive association between dysfunction (H+T for panel B and diabetic for panel C) and the single-cell co-expression of this gene with LINC00607.

LINC01013 overlaps with a SE but does not interact with any SE hubs in ECs at any time point. LINC02154 does not overlap with any SE.

5. *The endothelial cell dysfunction model warrants more support – particularly more physiological/pathophysiological and in vivo or ex vivo data. The glucose concentration is very high to be considered pathophysiological relevant in most disease setting. Confirmation and validation at lower concentrations or profiling of alternative stress models should be presented. Replication in arterial ECs is warranted. In the absence of in vivo experimental models, validation in ex vivo human EC samples would be of great value. Is this truly an EndoMT model – does the low level of SMA in HUVEC at day 7 (1.4% of cells) support this?*

We appreciate these comments. Indeed, blood-glucose concentrations are typically in the range of 4.9–6.9 mM for healthy patients. However, glucose levels up to 40 mM can be reached in diabetic patients (Wild *et al Diabetes Care*. 27:5–10, 2004). Notably, 25 mM glucose we used in *in vitro* cell culture experiments is commonly used in diabetes research and has been demonstrated in a large body of highly cited literature (e.g. Daille *et al. Diabetes* 54: 2179-2187, 2005; Katavetin *et al. J Am Soc Nephrol* 17: 1405-1413, 2006). In a recent paper, Wimmer *et al* used an even higher glucose concentration, i.e. 75 mM glucose, to mimic hyperglycemia *in vitro* (Wimmer *et al. Nature*. 565:505-510, 2019).

As the reviewer suggested, we have performed experiments with 15 and 20 mM HG, which also increased LINC00607 and SERPINE1, similar to that with 25 mM HG. These changes were attendant with the induction of monocyte chemoattractant protein 1 (MCP1, encoded by *CCL2*), another well-established inflammatory marker detected as a positive control.

LINC00607 and SERPINE1 induction by lower concentration of HG (high glucose). HUVECs were treated with high glucose (HG, 15 mM and 20 mM) respectively, plus TNF α (T) for 3 days. Controls (NM) were kept in medium containing the same concentration of mannitol. Data are presented as mean \pm SEM, n=5 in each group. * p <0.05 compared with NM control.

We have also performed new experiments in which we profiled and analyzed the single cell transcriptome in arterial ECs freshly isolated from healthy vs type 2 diabetic (T2D) donors using scRNA-seq. First of all, the top enriched disease term based on the pathway enrichment analysis is T2D from both comparisons. Second, there are over 200 common DEGs between the two comparisons, i.e. H+T vs NM ECs, and T2D vs health donor-derived ECs, including many well-identified EC dysfunction markers (e.g. CCL2, FN1, SERPINE1, EDN1, etc.) and various genes embedded in the emergent SE-enriched RNA-DNA interaction network (Figure to the right).

Furthermore, we have explored an alternative approach to validate our findings from *in vitro* using these new data from donor-derived ECs (as described in our response to Comment #4 and Fig. 5b, c in the revised manuscript). As detailed in the Results (Pages 15-17), both data from H+T-treated HUVECs and diabetic donor-derived arterial ECs revealed an increased co-expression of LINC00607 and SERPINE1 in the same single cells as compared to the respective control ECs. Compared to the other 29 genes embedded in the 12 Linc607SE-interacting SEs, which also show increased co-expression in the same single cells in ECs under diabetic conditions, SERPINE1 ranked as top 1 by odds ratio in both cultured ECs and donor-derived ECs.

Collectively, our data show multiple levels of consistence between our *in vitro* cultured EC model and arterial endothelium directly isolated from human donors.

To address whether our *in vitro* model is truly an EndoMT model, we referred to a recently published guideline article on EndoMT (Kovacic *et al. J Am Coll Cardiol* 73:190-209, 2019), a number of highly cited work demonstrating EndoMT *in vitro* including our own (Evrard *et al Nat Commun* 7:11853, 2016; He *et al. Circ Res* 120:354-365, 2017; Xiong, *et al Mol Cell* 69:689-698, 2018; Cooley *et al. Sci Transl Med* 6:227ra234, 2014; Dejana *et al Nat Commun* 8:14361, 2017), and personal communication with Dr. Michael Simons, an expert in EndoMT. The collected information supports that our model does represent EndoMT, evident by data at functional, cellular, and molecular levels. Specifically, we observed all the most characteristic changes of EndoMT including: 1) suppression of endothelial markers (e.g. eNOS) and induction

Heatmap of DE genes of the top 40 commonly upregulated between H+T treated ECs and T2D donor-derived arterial ECs. Cells were ordered by increasing SERPINE1 expression and binned per 25 cells.

of mesenchymal markers (e.g. SMA-alpha); and 2) morphological change as we demonstrated in Figure 1.

Regarding the low level of SMA in HUVEC at day 7 (1.4% of cells), we have two explanations: 1. Technically, scRNA-seq has limitations in detecting transcripts of low abundance. According to qPCR, α -SMA is highly induced in the HUVECs after H+T treatment but is still detected with a fairly high Cq (around 30) compared to genes abundantly expressed in ECs (Cq typically at 20-25). This relatively low level of expression may not be sufficiently captured by scRNA-seq; 2. EndoMT is believed to be a progressive transition, with a subpopulation of cells beginning to express α -SMA. Thus, although the detected percentage of cells expressing SMA at Day 7 is quite low (ie 1.4%), compared to 0.2% cells detected to express SMA in control ECs, the scRNA-seq still supports an induction of SMA in EndoMT.

6. The authors might elaborate more on the biological and translational significance and generalizability and potential impact of findings, in their concluding paragraph?

We are grateful for the reviewer's suggestion. We have elaborated more on these points in the discussion and concluding paragraphs now (Pages 18-21).

Minor comments:

1. Why do the authors expect specific directionality in the assumption here; "more DE genes upregulated than downregulated by Day 7 of H+T treatment (Fig. 2c), we argued that the increased inter-chromosomal RNA-chromatin interactions, rather than the decreased intra-chromosomal interactions, would be more directly related to the transcriptional induction of genes promoting EC dysfunction."

This is an excellent question. Our previous argument is indeed questionable. We have removed this statement in the revised manuscript.

2. What is the experimental evidence to support the endothelial SEs dataset (ref 30, dbSUPER) selected for endothelial (and what cell types)?

The SEs curated in dbSUPER were based on H3K27ac ChIP-seq data (Hnisz, *et al.* "Super-enhancers in the control of cell identity and disease." *Cell* 155: 934-947, 2013).

3. How many lncRNA detectable in these HUVEC experiments with 10G (speaks to low sensitivity/missing data) – e.g., compared to bulk-seq?

scRNA-seq detected a total of 44 lncRNAs in HUVECs expressed in greater than 10% of the total number of HUVECs. Bulk RNA-seq detected a total of 702 lncRNAs expressed at greater than 10 total reads in HUVECs. There are 39 lncRNAs detected by both scRNA-seq and bulk RNA-seq.

4. Fig 1. CD31 not expressed on most HUVECs at baseline – this is

It is not clear what was asked. CD31 is expressed in HUVECs at baseline but the quality of our original immunostaining images was not ideal. We have switched to VE-cadherin (VE-cad), a more specific EC marker to replace the CD31 staining in the revised Fig. 1 (also see figure to the right).

Immunostaining of HUVECs. VE-cad (red) and α -SMA (green) were detected using immunofluorescent staining in HUVECs treated by H+T or as control (NM). Nuclei were stained using DRAQ5 (blue). Scale bar = 50 μ m.

5. Figure 1b- Negative control bars do not have an error bar? Why was control set as 1 for both N+M and NT?

(Please note this is now Figure 1f) We set the relative mRNA levels of given genes in respective control groups to be 1 and therefore the control groups do not have error bars. In order to compare the relative fold changes of respective genes between the two different stimuli, ie. H+T vs TGF β +IL1 β , we set the respective control as 1 for each experiment.

6. Fig 6b-c. Control data for lncRNA looks very noisy – please address.

We appreciate this comment. We apologize that the previous Fig. 6b-c may have caused confusion.

Because our main focus is on the regulatory functions through inter-chromosomal SE interactions, we include in the revised manuscript only the coverage plot of *Linc607SE* transcripts over the region surrounding *Serpine1SE*, and at 200 kb resolution [Figure below and now Supplementary Fig. 5a; Day 0 (red), Day 3 (green), Day 7 (blue)], which is more appropriate to present the signal over *Serpine1SE*, given that the SE sizes are in the order of kilobases. We have also added information to the plot such as the y-axis scale and labels, the SE track, and the chromosomal coordinates.

We have also included in Fig. 3b the iMARGI derived contact matrices between Chromosome 2 (195Mb-235Mb) and Chromosome 7 (80Mb-120Mb), from the RNA (rows) to DNA (columns) in Day 0 (left), 3 (middle), and 7 (right) ECs, with arrows indicating the intersection of *Linc607SE* and *Serpine1SE*. Together, these two plots show time-dependent increase of the RNA from *Linc607SE* interaction with DNA of *Serpine1SE*.

7. Page 13 “The inhibitory effect of *LINC00607* LNA1 in monocyte adhesion to ECs was consistent using peripheral blood-derived monocytes from 4 different donors (Fig. 7d).” Figure 7d only shows 3 donors, change 4 to 3 in text.

To clarify, monocytes isolated from 4 different donors were used in this experiment for quantification, and images obtained from 3 donors were shown as representative. We have revised the figure legend to avoid the confusion.

8. Fix apparent error in sentence. “However, at the molecular level, whereas the H+T suppressed *eNOS* and *ICAM1* mRNA levels could be almost completely reversed, those of *LINC00607*, *SERPINE1*, *FNI*, and *CTGF* could not be reversed. In fact, the induction of *LINC00607*, *SERPINE1*, and *FNI* continued despite the removal of detrimental stimuli (Supplemental Fig. 10b).” H+T increased *ICAM* mRNA levels not suppressed. Also, are there any statistics for this figure?

We thank the reviewer for identifying this error. We have corrected this sentence and added statistics for this figure (Supplementary Fig. 10b).

9. Labeling of some figures should be improved: Figure 6a is missing Day 0, 3 and 7 above SE hub maps – can this figure be made bigger: Authors should refer to the names used in the figures for ease of reading e.g., *ACTA2* in figure 2E - add *ACTA2* in brackets after α -SMA: Also *eNOS*/*NOS3* does not appear in the heatmap of figure 2D; Clarify which figure is referred to here “Page 11, line 3 ‘Given this definition, there was only 1 SE hub (which overlaps with *MALAT1*) identified in control (Day 0) ECs, with 131 active SE nodes and 130 interacting SE pairs.’”

We have improved the labeling in the original Fig. 6a and made this figure larger (now Fig. 3a).

We have revised the original Fig. 2d, e accordingly. Please refer to the new Fig. 1, d, e. The sentence has been revised and a similar sentence “The only hub SE in control (Day 0) overlapped with the *MALAT1* gene” refers to the revised Fig. 3a.

10. *The authors should define the statistics used and definitions for *, **, *** in each of their figure legends? They should also include “n” numbers and experimental replicates for each experiment in every figure legend.*

We have now indicated the number of experiments and experimental replicates for each experiment in every figure legend.

11. *The authors should provide a figure to show expression of LINC00607 in the cytoplasm and nucleus – both by fractionation and smFISH?*

Accordingly, we have performed fractionation and RNA FISH to show the expression of LINC00607 in the cytoplasm and nucleus. We have also tried to perform smRNA FISH but unfortunately the Stellaris probes did not yield specific signals. We have included these data in Supplementary Fig. 6.

Localization of LINC00607.

HUVECs were treated as control (NM) or with H+T (HT) for 3 days. (Left) qPCR of LINC00607 plotted as percentages in subcellular fractions. Data are represented as mean \pm SEM from 5 replicas. (Right) RNA FISH of LINC00607 with a 5'FAM-labeled LNA probe (QIAGEN) (green). Nuclei were stained with DRAQ5. Scale bar = 50 μ m.

Reviewer #2 (Remarks to the Author):

In this highly innovative paper, Calandrelli and colleagues used a newly developed technology (called iMARGI) to study the RNA-chromatin interaction landscape changes during endothelial to mesenchymal transition (EndoMT) and impact on gene regulation. They found dramatic changes mainly characterized by marked increase of inter-chromosomal interactions and formation of new chromatin hubs. Interestingly, the Hi-C profiles show little changes. A key observation is that these hubs are enriched with super-enhancers, indicating the dynamic landscape of RNA-chromatin interactions may play a major role in gene regulation. They further showed that lncRNAs (with SERPINE1 as an example) could be a major player driving the RNA-chromatin dynamics. Altogether, the paper has discovered a novel gene regulatory mechanism through lncRNA mediated inter-chromosomal interactions which may have important consequences in diseases. The data are convincing and the paper is very well-written.

We are extremely grateful for the reviewer's positive comments.

I have a few minor questions/suggestions for the authors.

1. In Hi-C experiments, it is known to be difficult to identify inter-chromosomal interactions. I wonder if this is also the case here, and if so is there any way to determine the robustness of chromatin hub identification.

Of note, the SE hubs defined in this study are iMARGI derived, not Hi-C derived. Thus, the chromatin hub identification relies largely on the capability of iMARGI in detecting inter-chromosomal RNA-DNA interactions. Compared to Hi-C data, which showed less than 10% of the total read pairs as inter-chromosomal interactions (Fig. 2a), iMARGI read pairs comprised a much higher proportion of inter-chromosomal interactions, i.e. over 30% of the total read pairs in sample at Day 0, increasing over 60% in H+T samples at Day 3 and Day 7 (Fig. 2b). As we mentioned in the discussion of the revised manuscript, a potential model for explaining the observed global changes in the RNA-chromatin interactome could be a global "relaxation" of the globular structure of each chromosome, which obscures the boundaries of chromosome territories, thus making the transcripts from one chromosome more likely to diffuse to the spatial proximity of some genomic regions of another chromosome. This model of chromatin organization as a fractal globule is supported by the reverse linear relationship between the genomic interaction frequencies and distances. Furthermore, Hi-C data at days 3 and 7 exhibited a progressive decrease of DNA contact frequency at any given genomic distance (Supplementary Fig. 7a), as expected if this model was correct. As additional support, we also observed that TAD structures were weakened in dysfunctional ECs, with the progressive decrease of the proportion of reads mapped within TADs with H+T treatment (Supplementary Fig. 7b).

2. The model that lncRNAs mediated RNA-chromatin interactions could regulate gene expression by mediating SE formation is very interesting, but I think the argument could be made stronger if additional tests could be done to measure the degree of RNA-chromatin interactions changes due to lncRNA perturbation. Of particular interest is if the SE chromatin state is lost. The reported evidence here seems to be somewhat indirect.

We apologize that we did not explain our model clearly enough. We did not intend to say that lncRNA plays a role in SE formation or regulation of SE chromatin state. Instead, our data specifically suggest that lncRNA-SE interaction regulates the expression of critical genes in EC dysfunction.

To answer whether SE chromatin state is affected by lncRNA perturbation, we have performed ChIP-qPCR in HUVECs with lncRNA (e.g. LINC00607) knock-down (KD). Based on 3 independent experiments, we did not observe a consistent and significant effect caused by lncRNA KD in the SE chromatin state of several marker genes for EC dysfunction and also embedded in the emergent SEs for RNA-DNA interactions.

Effect of LINC00607 KD in H3K27ac. HUVECs were transfected with scramble (scr) or LNA1 inhibiting LINC00607 before subjected to mannitol (NM) or H+T (HT) as indicated. H3K27ac-marked regions near respective genes (based on Encode data from HUVECs) were selected for H3K27ac ChIP-qPCR, normalized by signals from input DNA. Data represent mean±SEM from 3 independent experiments.

3. *The single-cell RNAseq data does not seem to be fully utilized in the paper. The authors might consider to correlate the variation of lncRNA and target gene expression levels which would offer additional support to their model. i.e, the expression of a few key lncRNAs is required for activating SE associated genes.*

Taking the reviewer's excellent suggestion, we have correlated the LINC00607 and SERPINE1, as well as all the other genes contained in the LINC00607 interacting SEs emerging in H+T ECs using the scRNA-seq data from HUVECs. Furthermore, we have also performed scRNA-seq in arterial ECs derived from healthy vs diabetic donors and conducted similar correlation analysis to determine the co-expression of LINC00607 and these SE-associated genes. These data revealed that diabetic conditions induced consistent single-cell co-expression changes of LINC00607 and the target genes in our in vitro model (with H+T treatment) and human donor-derived ECs (from the diabetic individuals). These new exciting data have now been included in the new Fig. 5 (also see below).

Increased single-cell co-expression of LINC00607 and SERPINE1 in dysfunctional vascular endothelium. (A) H+T induced interactions (edges) between Linc607SE (yellow node) and hub SEs (red nodes). (B-C) Odds ratio (y axis) between single-cell co-expression levels and the health status of ECs, including H+T vs. control ECs (B) and diabetic (T2D) vs. healthy vascular endothelium (C). Each dot corresponds to a gene in any Linc607SE-interacting SE (panel A). A large odds ratio corresponds to a positive association between dysfunction (H+T for panel B and diabetic for panel C) and the single-cell co-expression of this gene with LINC00607.

Also, the tSNE plot (Figure 2b) seems to suggest that the cell states at the three time points are completely different. Would it be possible that some of these differences may be due to technical variation such as batch effects?

To answer this question, we have now included the t-SNE plot with all the samples separated. As visualized in the figure to the right, while cells at the three time points are totally separated in the t-SNE plot, cells from each biological replicate overlap almost completely. Therefore, the separation of samples from the three time points is unlikely due to batch effects. We have replaced the original Figure 2b with this figure (now in the new Fig. 1b).

4. In a broader context, the concept that chromatin hub plays a role in SE is not new, for example, see Huang et al. Nat Commun. 2018 Mar 5;9(1):943. It would be nice to acknowledge such previous work.

We have cited this excellent previous work in our revised manuscript, in the section “Hubs of inter-chromosomal RNA-DNA interaction networks” (page 12).

5. A list of SE hubs should be provided as a supplementary table.

We have provided the list of SE hubs, with embedded genes in the Supplementary Table 3.

Reviewer #3 (Remarks to the Author):

Calandrelli et al utilize a technique (iMARGI) that can map global RNA-genome interactions to understand how endothelial cells respond to stimuli that induce inflammation and endothelial-to-mesenchymal transition (endoMT). Surprisingly, global DNA-DNA interactions, as mapped by HiC did not change following exposure of endothelial cells to glucose and TNF-alpha for 7 days, despite major changes in cellular phenotype and gene expression programs (assessed by single-cell RNA-seq). Instead, they found that RNA-DNA interactions were highly altered, especially inter-chromosomal interactions. Many of these interactions appeared to preferentially occur in super-enhancer loci. While intra-chromosomal interactions decreased, inter-chromosomal interactions increased with glucose + TNF-alpha treatment. They examined one of these inter-chromosomal loci in detail (SERPINE1-LINC00607) and found that the lncRNA regulates SERPINE1 expression and several other inflammatory/endoMT genes as well as the inflammatory and endoMT phenotype of endothelial cells. The findings presented in this manuscript are highly novel and interesting. The data will be a rich resource of data for the community. However, several questions remain.

We are extremely grateful for the reviewer's highly positive feedback and have tried our best to address the remaining questions as detailed below.

Major Comments:

1) *The discovery of new inter-chromosomal hubs of RNA-DNA interactions at super-enhancer loci in response to glucose/TNF-alpha stimulation, including between the SERPINE1-LINC00607 loci, is intriguing. However, additional functional data connecting LINC00607 to the regulation of the hub(s) that it interacts with is required. Does knock-down of LINC00607 affect the proximity of the SERPINE1 locus and the LINC00607 locus? Are the genes dysregulated by LINC00607 knock-down preferentially in the areas of interaction between LINC00607 and the genome? This could be assessed by RNA-seq. The current manuscript only has a few select genes assessed by qRT-PCR. Does LINC00607 regulate the establishment of enhancer marks in the genomic regions that it interacts with?*

We appreciate these excellent comments. We have provided more functional data connecting LINC00607 to the regulation of 12 SEs that it interacts with: 1) We have performed RNA-seq using ECs with LINC00607 knockdown. The data revealed that in addition to SERPINE1, many genes that shown to be positively regulated by SERPINE1 were also downregulated by LINC00607 knock-down (new Fig. 4d). Furthermore, the expression of 13 other genes contained in 12 SEs was also suppressed by LINC00607 knockdown (see figure to the right and also new

RNA-seq data from ECs with LINC00607 KD. HUVECs were transfected with scramble (scr) or LNA inhibiting LINC00607 before subjected to mannitol (NM) or H+T (HT) as indicated. Heatmap plotted with Z score.

Supplementary Table 5). However, we are not able to determine whether “the genes dysregulated by LINC00607 knock-down preferentially in the areas of interaction between LINC00607 and the genome” because out of 1855 DE genes (1026 up and 829 down), many are not encoded in the genomic regions showing interactions with LINC00607. This is likely due to the secondary or indirect effects of LINC00607 knock-down.

2) We also took another approach to support the emergent regulatory link between LINC00607 and genes contained in all the Linc607SE-interacting SEs using scRNA-seq data from H+T-treated ECs as well as diabetic donor-derived ECs. The correlation analyses suggest that the SEs exhibiting H+T-induced interactions are more likely to co-express their harbored genes in the same single cells in dysfunctional ECs than in healthy ECs. Furthermore, the data from cultured ECs and donors’ ECs show a high degree of consistency. These new data are included in the new Fig. 5.

In regard to the establishment of enhancer marks, the SE states in ECs are already established before the HT treatment, according to Encode data and a previous report (Brown *et al Mol cell* 56: 219-231, 2014). To directly answer the reviewer’s question, we also performed H3K27ac ChIP in ECs with LINC00607 KD and detected the H3K27ac enrichment in several enhancer regions upstream of several EC dysfunction marker genes. Data from three biological replicates suggest that there is no consistent effect in SE state affected by LINC00607 inhibition (Figure below).

Effect of LINC00607 KD in H3K27ac. HUVECs were transfected with scramble (scr) or LNA1 inhibiting LINC00607 before subjected to mannitol (NM) or H+T (HT) as indicated. H3K27ac-marked regions near respective genes (based on Encode data from HUVECs) were selected for H3K27ac ChIP-qPCR, normalized by signals from input DNA. Data represent mean±SEM from 3 independent experiments.

As to whether knock-down of LINC00607 affects the proximity of the SERPINE1 locus and the LINC00607 locus, this is an outstanding question, but we currently do not have data to answer this question. Due to the refined focus of this study on addressing the role of caRNAs and RNA-chromatin contacts in transcriptional outcome, but not genomic interactions, we have moved the DNA FISH data suggesting the shortened distances between *Linc00607SE* and *Serpine1SE* in H+T-treated ECs to Supplementary Fig. 8 and only mentioned it in the Discussion.

2) It would be helpful to include additional lncRNA-DNA interaction maps for comparison to LINC00607. This would emphasize that the interaction with the SERPINE1 locus is unique for

this lncRNA. Is the interaction between LINC00607 RNA and the SERPINE1 locus in Fig. 6C significantly above background?

We appreciate these interesting comments from the reviewer. To clarify, we do not intend to emphasize that the interaction with the *SERPINE1* locus is unique for this lncRNA. The reasons are two-fold:

- 1) Rationale: to test our model in which inter-chromosomal RNA-chromatin interactions activate critical genes in the target genomic regions to contribute to EC dysfunction, a typical approach would be to inhibit the “upstream” transcripts and examine the “downstream” critical gene expression. We reasoned that if we perturb a mRNA, we would not be able to dissect the regulatory role of this mRNA from the functional role of its encoded protein. On the other hand, if we perturb a non-coding RNA (ncRNA), we can largely attribute the downstream effect to the ncRNA *per se*. This rationale led us to perturb the RNA-chromatin interaction between two SEs by inhibiting the ncRNA transcribed from a SE. Among all the H+T-induced interacting SEs, 26 SEs do not contain any coding gene and contain at least one lncRNA gene. Among the lncRNAs contained in these SEs, only *Linc00607* was reported as a H+T-induced lncRNA by scRNA-seq analysis at the statistical threshold of FDR < 1e-32. Therefore, we selected LINC00607 in the perturbation experiment. Likewise, we selected *SERPINE1* as a target gene because 1) it is embedded in the *Serpine1SE*, an emergent hub SE in H+T-treated ECs; 2) The *SERPINE1* gene encodes PAI-1, a key regulator of EC dysfunction linking hyperglycemia, inflammation, EndoMT; and 3) *SERPINE1* is also one of the top H+T-induced genes in ECs identified by scRNA-seq.
- 2) Results: In addition to *SERPINE1*, inhibition of LINC00607 also leads to suppression of several other genes embedded in the SEs that exhibited increased interactions with LINC00607 under H+T treatment (please refer to the new Supplementary Table 4). In line with this data, the correlation analysis using scRNA-seq data from H+T-treated ECs and diabetic donor-derived ECs also showed an increase in the odds of co-expressing LINC00607 and these genes in the same single cells (see new Fig. 5). Likewise, there are other lncRNA genes contained in the emergent hub SEs that form increased RNA-chromatin interactions under H+T treatment (Supplementary Table 3). The lncRNAs derived from these regions may also be involved in the transcriptional induction of genes contributing to EC dysfunction.

We have also added these points in the revised Results (on Page 13) and Discussion (on Page 19).

In order to demonstrate that the interaction in previous Fig. 6B-C between *Linc607SE* and *Serpine1SE* was above background, we have performed peak calling using HOMER (<http://homer.ucsd.edu/homer/index.html>). Peaks were called on the read coverage tracks of Day 7 (blue), using a peak size of 100 kb (approximately the average size of a super enhancer). Coverage plots at 1 Mb resolution were re-plotted adding also the peak track (black dashed). As shown in the figures below, the algorithm not only called a significant peak over *Serpine1SE*, but that peak is also the one with the highest score across the entire chromosome 7. Therefore, indeed, *Linc607SE-Serpine1SE* locus was significantly above background.

Coverage plot of *Linc607SE* RNA transcripts over chromosome 7. Y-axis is “read pairs/total pairs * 10^8 ” for the read coverage tracks (Day 0, red; Day 3, green; Day 7, blue). The peak track (black dashed) represents the peak score, scaled by a factor used for visualization purpose only, in order to reduce the overlapping with coverage tracks and allow for better visualization of the peaks.

3) *The imaging analysis of the endoMT phenotype could be improved. Confocal imaging of VE-Cadherin and co-staining with Phalloidin (to show cell shape change) would be helpful. The brightfield images provided are difficult to interpret and the CD31 staining was very patchy.*

To improve the imaging analysis of the EC phenotypic change, we have performed co-staining of VE-cad (a more specific marker for ECs than CD31) and α -SMA (most commonly used EndoMT marker) to generate images with higher quality. Moreover, we have also taken the reviewer’s suggestion and performed co-staining of VE-cad and phalloidin separately from α -SMA (due to the overlapping fluorescent signal). The new data are now included in Fig. 1g.

4) *Are genomic DNA-DNA interactions between the *SERPINE1* and *LINC00607* loci observable from HiC data and do they change during the time-course of treatment?*

Hi-C data do not show any significant *DNA-DNA interactions between the *SERPINE1* and *LINC00607* loci*, neither any remarkable changes during the time course. We have included this data in the new Fig. 3c (also included below).

5) Is there an independent method that can be used to validate some of the RNA-DNA interactions? The finding of such widespread inter-chromosomal interactions is very intriguing, but additional independent evidence would be helpful. This is especially the case if only one replicate was used for analysis. The number of replicates should be clearly stated in the paper. Does LINC00607 RNA FISH reveal co-localization with the SERPINE1 locus?

We have repeated the iMARGI experiment and generated a second iMARGI dataset from Day 0 and Day 7 ECs. Based on these replicates, we have added error bars to every quantification figure related to these replicates, namely Fig. 2, b and e and Supplementary Fig. 4, a, b and f. In the results, we now describe two types of robustness analyses based on (1) comparing the H+T-induced changes between Day 7 and Day 0 each with two biological replicates, and (2) merging and splitting the two time points after treatment (Day 3, Day 7) for comparison to Day 0. We have also provided a table to list the numbers of replicates used in each high-through profiling experiments (Supplementary Fig. 2).

We have also tried to use an independent method, i.e. ChIRP to validate the increased *Linc607SE-Serpine1SE* RNA-DNA interactions by H+T. The data below suggest that there is a moderate increase in LINC00607 association with one of the two selective regions (based on peaks identified in iMARGI) of *Serpine1* SE in both replicates. However, the data are not as reproducible as those in iMARGI.

LINC00607 ChIRP-qPCR for SERPINE1. Twenty ChIRP probes complementary to LINC00607 were designed and synthesized (Stellaris). ChIRP-qPCR was performed as in Miao *et al. Nat Commun* 9: 292, 2018. Interaction between LINC00607 and the genomic locus of β -actin was detected as a negative control, which did not show significant binding.

6) *Is LINC00607 enriched in the nucleus? This data should be available from the raw data used in Fig. 7B.*

Yes, LINC00607 is enriched in the nucleus, as shown in the figure below. We have also included this in the Supplementary Figure 6d.

qPCR of LINC00607 in subcellular fractions. HUVECs were treated with 25 mM mannitol (NM) or high glucose (25 mM) + TNF α (HT) for 3 days, subcellular fractionation was performed as described in Miao et al. Nat Commun 9: 292, 2018. Data are plotted as percentages in each fraction, with mean \pm SEM from 5 replicas.

7) *The data should be made available in a public data repository.*

We have made our data available at GEO with accession number GSE135357.

REVIEWERS' COMMENTS:

Reviewer #2 (Remarks to the Author):

The authors have satisfactorily addressed all my concerns.

Reviewer #3 (Remarks to the Author):

The authors have responded to my previous comments. This is a highly novel study. I just have two minor comments.

- 1) Many of the figure panels are highly pixelated. Resolution needs to be improved.
- 2) Fig. 3B and C – Labelling these as 'RNA-DNA and DNA-DNA' directly on the figure rather than in the legend would improve the clarity of the figure.

Point-to-point response

Reviewer 3:

1) Many of the figure panels are highly pixelated. Resolution needs to be improved.

We have checked and improved the resolution of every figure.

2) Fig. 3B and C – Labelling these as ‘RNA-DNA and DNA-DNA’ directly on the figure rather than in the legend would improve the clarity of the figure.

We have labeled ‘RNA-DNA and DNA-DNA’ directly on Fig. 3.